# Token-Sparse Medical Multimodal Reasoning via Dual-Stream Reinforcement Learning

**Kaitao Chen** [1 2]   **Weiqian Zhao** [3]   **Jiamin Wu** [2]   **Qihao Zheng** [2]   **Shangquan Sun** [2]   **Chunfeng Song** [2]
**Xiaosong Wang** [2]   **Mu Zhou** [4]   **Mianxin Liu** [2]

## Abstract

Vision-language models (VLMs) combining reinforcement learning (RL) ignite remarkable progress in multimodal reasoning, yet still struggle with medical images, which typically exhibit extremely sparse visual evidence to inform clinical decision-making. We recognize that pruning visual tokens outside the grounding region greatly enhances medical reasoning. However, a united RL framework for active visual token pruning (VTP) and medical multimodal reasoning remains unestablished. Here, we propose a dual-stream RL framework, **ViToS**, to fulfill token pruning and question answering. ViToS trains one policy model with two task branches, where one focuses on grounding while the other conducts token-sparse reasoning after VTP. Furthermore, we solve the coupled policy learning problem by introducing the cross-feedback sequential optimization, avoiding gradient conflict and facilitating convergence of the shared policy model. Evaluated on seven medical benchmarks, our method reduces visual tokens to 77% of the original sequence length while achieving a 108.27% relative performance on Lingshu-7B and 104.16% relative performance on HuatuoGPT-Vision-7B. Overall, ViToS delivers superior performance and inference speedup, establishing an efficient paradigm for medical multimodal reasoning.

## 1. Introduction

Vision-language models (VLMs) have achieved remarkable progress in multimodal tasks including visual question answering (VQA) and cross-modal retrieval (Zhang et al.,

---
[1]Fudan University [2]Shanghai Artificial Intelligence Laboratory [3]Shanghai Jiao Tong University [4]Rutgers University. Correspondence to: Mianxin Liu <liumianxin@pjlab.org.cn>.

*Proceedings of the 43[rd] International Conference on Machine Learning*, Seoul, South Korea. PMLR 306, 2026. Copyright 2026 by the author(s).

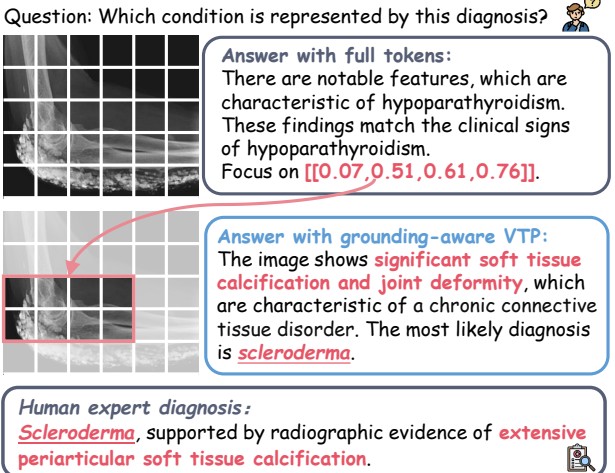

Question: Which condition is represented by this diagnosis?

**Answer with full tokens:**
There are notable features, which are characteristic of hypoparathyroidism. These findings match the clinical signs of hypoparathyroidism.
Focus on [[0.07,0.51,0.61,0.76]].

**Answer with grounding-aware VTP:**
The image shows significant soft tissue calcification and joint deformity, which are characteristic of a chronic connective tissue disorder. The most likely diagnosis is scleroderma.

*Human expert diagnosis:*
Scleroderma, supported by radiographic evidence of extensive periarticular soft tissue calcification.

*Figure 1.* Illustration of grounding-aware VTP. By focusing its reasoning on grounded tokens, the model correctly diagnoses scleroderma, consistent with the human expert assessment.

2024b; Xu et al., 2025b; Chen et al., 2024a; Jiang et al., 2025). These VLMs typically adopt a uniform visual token encoder that maps images into dense visual tokens for large language model (LLM) decoding, therefore introducing substantial visual token redundancy (Bai et al., 2025a; Zhu et al., 2025; Liu et al., 2023). This redundancy escalates computational overhead and produces distracting patterns (Guo et al., 2025b; Chen et al., 2024b; Zhang et al., 2024a), ultimately impairing the model's ability to focus on the critical visual evidence. This phenomenon is especially prominent in healthcare applications, where images contain rich information, yet only extremely sparse visual evidence is relevant for clinical decision-making (Chen et al., 2025a).

To tackle the distracting pattern arising from the visual token redundancy, we conduct preliminary experiments on several medical VQA benchmarks using a visual token pruning (VTP) strategy, as illustrated in Figure 1. In particular, our grounding-aware VTP strategy selects grounded token and fuses the remaining tokens to the grounded ones based on their semantic similarity. As seen in Figure 2, our grounding-aware VTP strategy improves VQA perfor-

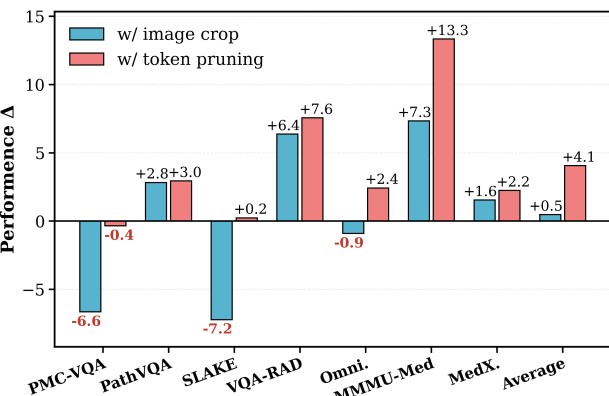

*Figure 2.* Impact of grounding-aware VTP and image cropping on performance across seven medical VQA benchmarks.

mance, yielding an average gain of 4.1% on Lingshu-7B (Xu et al., 2025b). Note that background information can be retained in reserved tokens via self-attention in the visual encoder (Dosovitskiy et al., 2020), as well as through the token fusion strategy we employ. In contrast, traditional image cropping, a common redundancy-reduction tactic, leads to severe performance fluctuations due to the indiscriminate removal of background details. This contrast underscores that focusing on grounded tokens via VTP can more effectively mitigate visual distraction and boost reasoning ability. Nevertheless, this improvement hinges on visual pruning at inference time, rather than being internalized within the model. Toward efficient medical visual reasoning with sparse evidence, we must address a fundamental question: ***What training scheme can effectively optimize visual token-sparse reasoning ability in medical VLMs?***

In this study, we propose a novel dual-stream reinforcement learning (RL) framework for medical **Vi**sual **To**ken-**S**parse reasoning, named **ViToS**[1]. ViToS not only reformulates VTP as a policy-guided evidence selection process but also facilitates end-to-end training for token-sparse medical multimodal reasoning. ViToS trains one policy model with two cascaded branches. First, the localization branch generates spatial grounding for VTP. Second, a token-sparse reasoning branch employs the proposed grounding-aware VTP scheme and complete token-sparse reasoning. Furthermore, we develop the cross-feedback sequential optimization to solve the coupled policy learning problem emerged in the dual-stream RL, avoiding gradient conflict and facilitating convergence of the shared policy model.

Extensive experiments across seven benchmarks demonstrate that ViToS consistently yields substantial performance gains. Our method reduces visual tokens to 77% of the original sequence length while achieving 108.27% relative

---

[1]Code is available at https://github.com/JLINEkai/ViToS.

performance on Lingshu-7B (Xu et al., 2025b) and 104.16% relative performance on HuatuoGPT-Vision-7B (Chen et al., 2024a). Compared with strong VTP methods (Zhang et al., 2025a; Xing et al., 2024; Zhang et al., 2025a), our approach achieves substantial performance gains while also providing an appealing inference speedup with support from the vLLM engine (Kwon et al., 2023).

Our main contributions are summarized as follows:

- We propose a dual-stream RL framework using a united policy model for grounding and token-sparse reasoning, with a cross-feedback sequential optimization scheme to address coupled policy learning.

- We realize a trainable grounding-aware VTP with RL, reformulating VTP as a policy-guided evidence selection process rather than a heuristic token reduction.

- Compared with strong medical VLMs and VTP methods, extensive experiments demonstrate that our method consistently yields substantial performance gains while providing a inference speedup.

## 2. Related Work

**Medical Vision-Language Models.** Vision–language models (VLMs) have demonstrated strong capabilities in handling multimodal data, leading to the emergence of advanced medical VLMs, such as LLaVA-Med (Li et al., 2023), HuatuoGPT-Vision (Chen et al., 2024a), Lingshu (Xu et al., 2025b), and Hulu-Med (Jiang et al., 2025). These models are typically trained via medical image-text alignment and multimodal instruction finetuning. To emphasize fine-grained visual evidence, R-LLaVA (Chen et al., 2025b) and MedTrinity (Xie et al., 2024) introduce structured region-of-interest (RoI) supervision to focus on the critical region. Despite these advances, existing medical VLMs primarily operate at the image level, where reasoning is still conducted over dense and uniform visual token representations. To address this hurdle, we explicitly target visual token selection, enabling faithful medical reasoning under compact and selectively compressed visual tokens.

**Reinforcement Learning for Medical VLMs.** Inspired by the model superiority on reinforcement learning (e.g., Deepseek-R1 (Guo et al., 2025a)), researchers applied the RL training paradigms on medical VLMs (Rui et al., 2025; Zhang et al., 2025b). Distinguished examples are Med-R1 (Lai et al., 2025) and MedVLM-R1 (Pan et al., 2025) that design verified reward signals to optimize models' reasoning ability. MedGround-R1 (Xu et al., 2025a) guides the model to concentrate on key regions using chain-of-box template. To address the inherent problem of distracting parts in medical images, ViTAR (Chen et al., 2025a) constructs the cognitive chain to mimic the behavior of human doctors when diagnosing medical conditions. We recognize that

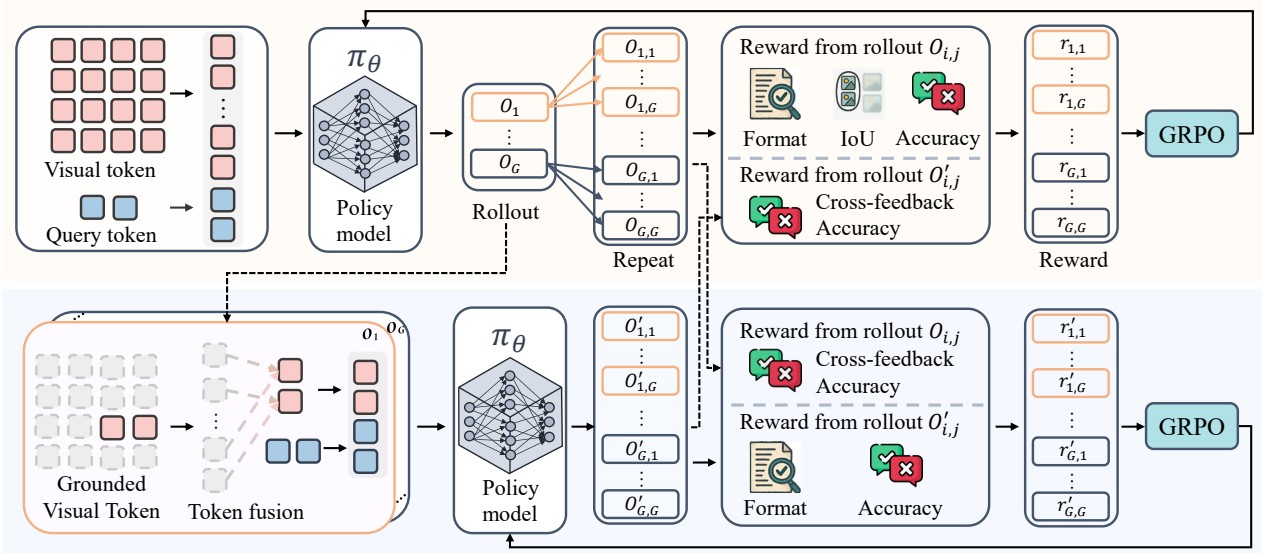

*Figure 3.* Overview of the proposed dual-stream RL framework with a unified policy model. The localization branch (top) identifies where to focus, and token-sparse reasoning branch (bottom) focuses on how to reason over compressed tokens. Two branches are sequentially optimized through reciprocal cross-feedback rewards, where each branch provides reinforcement signals to guide the other.

prior RL-based approaches mainly enhance one ability by a single RL optimization branch. Unlike these approaches, we investigate a dual-stream RL strategy that jointly internalizes grounding-aware VTP and token-sparse reasoning abilities of one policy model in an unified RL training, by formulating visual evidence selection and sparse evidence reasoning as a coupled policy learning problem.

**Visual Token Pruning.** Redundant VLM-generated visual token heavily aggravates computational cost and distracts reasoning (Bai et al., 2025a). Visual token pruning (VTP) selects useful visual tokens while maintaining overall model performance. Representative approaches are TopV (Yang et al., 2025a), A-ViT (Yin et al., 2021) and FlowCut (Tong et al., 2025) that identify salient visual tokens by calculating attention scores or similarity metrics. These methods only concentrate on semantic relevance and neglect global information. Growing studies merge redundant tokens into several critical tokens, such as SparseVLM (Zhang et al., 2024a) and LLaVA-PruMerge (Shang et al., 2025). Yet they cause the destruction of positional information. As a result, IVC-Prune (Sun et al., 2026) recognizes that implicit visual coordinates are essential for spatial reasoning and foreground tokens. GAP (Chien et al., 2025) adjusts the remained tokens' position embeddings after pruning. Note that above VTP methods operate independently and receive no optimization through end-to-end training for domain-specific reasoning. Although VisionThink (Yang et al., 2025c) applies RL and reduces the number of visual tokens via image downsampling, the role of RL is limited to deciding whether the downsampled image is sufficient for reasoning. By contrast, our work reframes VTP as a

form of selective visual evidence and reasoning process via end-to-end training, shifting the focus from pure efficiency to achieving medical reasoning with compact visual tokens.

## 3. Method

### 3.1. Dual-Stream Reinforcement Learning (DS-RL)

In Figure 3, we propose a dual-stream RL framework including localization branch and token-sparse reasoning branch. The localization branch focuses on proposing which spatial regions to preserve visual tokens, whereas the token-sparse reasoning branch focuses on how to reason over compressed visual tokens through VTP.

**Localization Branch.** We treat the VLM as an optimizable policy model $\pi_\theta$. Given a instruction $X = \{x_v, x_t\}$, where $x_v$ denotes the image input and $x_t$ the textual query, the policy $\pi_\theta$ generates a group rollout $\{o_1, o_2, \ldots, o_G\}$ following a structured format, including `reasoning`, `coordinate`, and `output`. Here, `reasoning` represents the trajectory of linguistic reasoning, while `output` is the final prediction. `Coordinate` corresponds to the predicted foreground bounding box $b_{\text{pred}}$.

**Token-Sparse Reasoning Branch.** We utility grounding-aware visual token pruning (GTP) in this branch. The visual encoder of the policy $\pi_\theta$ first encodes input images into a sequence of visual tokens $\mathbf{E} \in \mathbb{R}^{N \times D}$, where $N$ denotes the number of tokens and $D$ the feature dimension. For each output $o_g$ ($g \in [0, 1, \ldots, G]$) from the first branch, we obtain a corresponding predicted foreground bounding box $b_{\text{pred}}^{(g)}$.

Conditioned on $b_{\text{pred}}^{(g)}$, the visual tokens $\mathbf{E}$ are partitioned into two subsets: foreground tokens $\mathbf{E}_{\text{kept}}^{(g)}$ located inside the bounding box, and background tokens $\mathbf{E}_{\text{others}}^{(g)}$ outside the box.

To compress background information, we compute a normalized cosine similarity between each background token $e_j \in \mathbf{E}_{\text{others}}^{(g)}$ and all foreground tokens in $\mathbf{E}_{\text{kept}}^{(g)}$. Each background token is then assigned to its most semantically similar foreground token and fused through weighted aggregation. This token pruning and fusion process yields a sparse yet informative visual representation $\hat{\mathbf{E}}_{\text{kept}}^{(g)}$, where global contextual cues are merged into foreground-centric tokens. Through this mechanism, the number of visual tokens is reduced from $N$ to $|\mathbf{E}_{\text{kept}}^{(g)}|$, while retaining both fine-grained foreground details and global semantics.

**Token-Sparse Reasoning with Group Rollout.** The above token pruning procedure is independently applied to all $G$ first-branch outputs, resulting in a set of sparse visual representations $\{\hat{\mathbf{E}}_{\text{kept}}^{(1)}, \hat{\mathbf{E}}_{\text{kept}}^{(2)}, \ldots, \hat{\mathbf{E}}_{\text{kept}}^{(G)}\}$. Since token pruning disrupts the original spatial ordering and invalidates positional embeddings, we reassign positional information using rotary positional embeddings (Bai et al., 2025b), which preserve relative relationships among the foreground tokens.

Each sparse visual representation $\hat{\mathbf{E}}_{\text{kept}}^{(g)}$ is then fed into the decoder of second branch to perform token-sparse reasoning. We conduct group rollouts for each $\hat{\mathbf{E}}_{\text{kept}}^{(g)}$, generating $G$ reasoning trajectories. Finally, this results in an expanded set of outputs $\{o'_{g,1}, o'_{g,2}, \ldots, o'_{g,G}\}_{g=1}^{G}$. Here, $\{o'_{g,1}, o'_{g,2}, \ldots, o'_{g,G}\}$ denotes the group rollout conditioned on the candidate $o_g$ from first branch. By enforcing reasoning over compact and foreground-centric visual evidence, this branch effectively promotes faithful, token-sparse inference. In this branch, the generated response consists of `reasoning` and `output`, without `coordinate`.

**Rollout Alignment Across Branches.** To ensure consistent rollout between the localization branch and the token-sparse reasoning branch, we align the first-branch outputs with the expanded rollout space in second-branch outputs. Each localization output $o_g$ is repeated $G$ times, forming the aligned set $\{o_{g,1}, o_{g,2}, \ldots, o_{g,G}\}_{g=1}^{G}$. This expansion enables one-to-one correspondence between localization predictions and the token-sparse reasoning trajectories, allowing a unified reward computation.

## 3.2. Verifiable Reward Design

We adopt a composite reward system to evaluate the quality of sampled responses along three dimensions: format integrity, spatial precision, and diagnostic accuracy.

**Format Reward.** To ensure stable execution of the dual-branch rollout, the model is required to strictly adhere to pre-defined output schemas, earning a reward $R_{\text{format}} = 1$ only if the localization branch includes `<think>` `</think>`, `<bbox>` `</bbox>`, and `<answer>` `</answer>` tags in the correct order, and the sparse-evidence reasoning branch similarly contains the `<think>` `</think>` and `<answer>` `</answer>` tags; any deviation from these formatting requirements results in $R_{\text{format}} = 0$.

**Spatial Localization Reward.** This reward is applied exclusively to the localization branch to assess foreground localization quality. We compute the intersection-over-union (IoU) as $R_{\text{IoU}}$ between the predicted bounding box $b_{\text{pred}}$ and the ground-truth box $b_{\text{gt}}$. This signal provides direct optimization feedback for coordinate generation and serves as the foundation for accurate grounding-aware visual token pruning in token-sparse reasoning branch.

**Task-Specific Accuracy Reward.** We define the accuracy reward as a binary variable: $R_{\text{acc}} = 1$ if the predicted answer matches the ground-truth label, and $R_{\text{acc}} = 0$ otherwise.

## 3.3. Sequential Optimization with Cross-Feedback

Although the dual-branch rollout establishes a cascaded localization and reasoning relationship, directly optimizing both branches with independent supervision signals often leads to gradient conflicts and policy drift. Given the strong dependency of reasoning on precise visual evidence, we solve a coupled policy learning problem and propose a cross-feedback–driven sequential optimization strategy.

**Cross-Feedback Reward.** Inspired by Vision-SR1 (Li et al., 2025), we introduce the cross-feedback reward as reciprocal supervision signals between branches. For the localization branch, we use the accuracy reward $R_{\text{acc}}^S$ from token-sparse reasoning branch as a cross-feedback signal. The final aggregated reward $R^L$ for localization branch is defined as:

$$R_{\text{total}}^L = 3 \cdot (1 - \lambda) \cdot R_{\text{format}}^L + \lambda \cdot R_{\text{IoU}}^L \\ + \lambda \cdot R_{\text{acc}}^L + \underbrace{\lambda \cdot R_{\text{acc}}^S}_{\text{cross-feedback}} . \quad (1)$$

Here, the superscripts of $R^L$ and $R^S$ are used to distinguish whether a reward term is derived from the localization branch (L) or the token-sparse reasoning branch (S), respectively. The coefficients $\lambda$ are non-negative scalar hyperparameters that control the relative contribution of each reward component. Intuitively, a high-quality localization prediction $b_{\text{pred}}$ should preserve sufficient discriminative evidence to support downstream reasoning. If token-sparse reasoning branch can still answer correctly when operating solely on the pruned tokens, it indicates that localization branch has successfully captured critical visual cues, and should therefore receive cross-feedback reward from $R_{\text{acc}}^S$.

For the token-sparse reasoning branch, we incorporate the

accuracy reward $R_{\text{acc}}^L$ from localization branch. The final aggregated reward is defined as:

$$R_{\text{total}}^S = 2 \cdot (1 - \lambda) \cdot R_{\text{format}}^S + \lambda \cdot R_{\text{acc}}^S + \underbrace{\lambda \cdot R_{\text{acc}}^L}_{\text{cross-feedback}} \cdot \quad (2)$$

This design mitigates training instability caused by operating on pruned visual tokens, and ensures that correct reasoning grounded in accurate localization receives a higher advantage. As a result, the model is encouraged to focus its attention on valid evidence regions selected by $b_{\text{pred}}$.

**Sequential Optimization.** Based on the above reward design, we adopt a sequential optimization strategy. First, we only optimize the localization branch via reward $R_{\text{total}}^L$ until a reliable localization capability is established. During this stage, the token-sparse reasoning branch is excluded from gradient computation. After the localization policy converges, the optimization focus is shifted to the token-sparse reasoning branch. In this stage, the model leverages the fixed localization policy to perform token selection and optimizes reasoning via reward $R_{\text{total}}^S$. The localization branch remains frozen, with gradient computation disabled throughout this stage. By explicit gradient decoupling between branches, we can optimize the a united policy $\pi_\theta$ by maximizing the following GRPO objective (Guo et al., 2025a):

$$\mathcal{L}_{\text{GRPO}}(\theta) = \mathbb{E}_{X \sim \mathcal{D}} \left[ \frac{1}{G} \sum_{g=1}^{G} \left( \min \left( \frac{\pi_\theta(o_g \mid X)}{\pi_{\theta_{\text{old}}}(o_g \mid X)} \hat{A}_g, \right. \right. \right.$$

$$\left. \left. \left. \text{clip} \left( \frac{\pi_\theta(o_g \mid X)}{\pi_{\theta_{\text{old}}}(o_g \mid X)}, 1 - \epsilon, 1 + \epsilon \right) \hat{A}_g \right) - \beta D_{\text{KL}}(\pi_\theta \| \pi_{\text{ref}}) \right) \right].$$
$$(3)$$

In this objective, the policy ratio $\frac{\pi_\theta}{\pi_{\theta_{\text{old}}}}$ measures the parameter shift between the current update and the sampling step. $\hat{A}_g$ is the advantage value by computing relative scores within the group reward. The hyperparameter $\epsilon$ determines the clipping range. The KL regularization term $D_{\text{KL}}(\pi_\theta \| \pi_{\text{ref}})$, weighted by the penalty coefficient $\beta$, constrains the policy to preserve the fundamental language generation capability of the original pretrained model $\pi_{\text{ref}}$.

Through this sequential optimization, the abilities of visual grounding and token-sparse reasoning are ultimately internalized within a single policy model $\pi_\theta$, enabling the model to autonomously attend to critical regions and generate reliable diagnostic predictions during inference.

## 4. Experiments

### 4.1. Evaluation Setup

**Benchmarks.** We evaluate our framework on a diverse suite of medical VQA benchmarks designed to cover both perceptual understanding and advanced clinical reasoning, using the same experimental settings as ViTAR (Chen et al.,

2025a). PathVQA (He et al., 2020), SLAKE (Liu et al., 2021), and VQA-RAD (Lau et al., 2018) primarily assess fine-grained medical perceptual understanding. OmniMed-VQA (Hu et al., 2024) reformulates heterogeneous medical image classification tasks into a unified VQA setting, emphasizing robust multimodal perception across modalities. PMC-VQA (Zhang et al., 2023) targets knowledge-intensive medical question answering, requiring the integration of visual evidence with medical knowledge. To evaluate higher-level reasoning, MMMU-Med, the Health & Medicine track of MMMU (Yue et al., 2024), introduces complex multimodal reasoning scenarios in the medical domain. MedX-pertQA (Zuo et al., 2025) simulates real clinical examination settings to assess expert-level medical decision-making.

**Implementation Details.** We validate our method on two representative 7B-scale medical VLMs, including Lingshu (Xu et al., 2025b) and HuatuoGPT-Vision (Chen et al., 2024a), and further evaluate scalability using a larger-scale Lingshu-32B. We compare our approach with representative visual token pruning methods, including VisionZip (Yang et al., 2025b), VScan (Zhang et al., 2025a), FastV (Chen et al., 2024b), and PDrop (Xing et al., 2024). Unlike our dynamic, policy-guided token pruning strategy, these methods rely on fixed heuristic pruning rules with static parameters. We compare our approach with strong medical RL methods, including Med-R1 (Lai et al., 2025), MedVLM-R1 (Pan et al., 2025), MedCCO (Rui et al., 2025), Med-VLThinker (Huang et al., 2025), and ViTAR (Chen et al., 2025a). Our method is trained on 8 NVIDIA H200 GPUs with 8 hours, using the veRL (Sheng et al., 2024) training framework. Grounding-aware VTP is enabled at inference time and is fully compatible with the vLLM (Kwon et al., 2023) inference framework. The ablation studies are completed using the default Lingshu-7B. Detailed experimental settings and training hyperparameter configurations are provided in the Appendix Section C and D.

### 4.2. Main Results

In Table 1, our framework consistently outperforms baselines across diverse model types (Lingshu and HuatuoGPT-Vision) and scales (7B and 32B). Applying grounding-aware visual token pruning (GTP) to Lingshu-7B results in a performance leap from 63.68% to 67.74%. Dual-stream reinforcement learning (DS-RL) bridges the gap on Lingshu-7B, boosting average performance by 3.77%. Combining DS-RL with GTP yields strong results, improving average accuracy by 5.27% on Lingshu-7B. This suggests that DS-RL and GTP function orthogonally to create a distinct synergy: the former aligns the model with medical reasoning logic during training, while the latter actively mitigates visual redundancy during inference. Furthermore, our approach demonstrates robust scalability. On the larger Lingshu-32B, combining DS-RL with GTP unlocks a remarkable potential,

*Table 1.* Main results on seven medical multimodal benchmarks. We compare the performance of our proposed pruning and RL strategies against the instruct models across different medical VLMs (Lingshu (Xu et al., 2025b) and HuatuoGPT-Vision (Chen et al., 2024a)) and scales (7B and 32B). The results demonstrate that our combined strategy consistently achieves the best performance.

| | PMC-VQA | PathVQA | SLAKE | VQA-RAD | Omni. | MMMU-Med | MedX. | Average |
|---|---|---|---|---|---|---|---|---|
| ***Lingshu-7B*** | | | | | | | | |
| Instruct Model | 60.20 | 71.98 | 81.25 | 64.14 | 81.34 | 63.33 | 23.50 | 63.68 |
| GTP | 59.85 | 74.93 | 81.49 | 71.71 | 83.77 | 76.67 | 25.75 | 67.74 |
| DS-RL | 61.65 | 77.16 | 82.45 | 69.32 | 80.57 | 76.00 | 25.00 | 67.45 |
| DS-RL + GTP | 62.40 | 77.19 | 84.62 | 71.31 | 83.30 | 78.00 | 25.80 | 68.95 |
| ***HuatuoGPT-Vision-7B*** | | | | | | | | |
| Instruct Model | 55.05 | 65.76 | 70.91 | 66.53 | 71.72 | 45.33 | 22.35 | 56.81 |
| GTP | 52.85 | 62.05 | 74.52 | 65.34 | 74.14 | 51.33 | 22.90 | 57.59 |
| DS-RL | 53.90 | 64.99 | 75.48 | 68.13 | 71.86 | 50.67 | 22.55 | 58.23 |
| DS-RL + GTP | 52.40 | 63.80 | 78.61 | 70.52 | 74.96 | 50.00 | 23.90 | 59.17 |
| ***Lingshu-32B*** | | | | | | | | |
| Instruct Model | 61.70 | 77.28 | 85.82 | 70.52 | 79.52 | 62.00 | 28.40 | 66.46 |
| GTP | 62.85 | 85.19 | 90.38 | 69.32 | 84.73 | 66.67 | 28.00 | 69.59 |
| DS-RL | 64.35 | 85.34 | 90.14 | 70.52 | 83.72 | 66.00 | 28.10 | 69.74 |
| DS-RL + GTP | 63.05 | 85.37 | 90.62 | 72.51 | 83.45 | 66.67 | 29.80 | 70.21 |

with SLAKE accuracy peaking at 90.62% and PathVQA peaking at 85.37%. This key finding points to a paradigm shift in medical VLMs, where effective reasoning is driven by selective visual evidence and reasoning-aligned training, rather than exhaustive processing of dense visual tokens.

### 4.3. Comparison with Previous Pruning Methods

From Table 2, our approach consistently outperforms representative visual token pruning methods across all evaluated benchmarks and both baseline VLMs. Under the pruned visual token, our approach consistently surpasses all competing pruning methods. On Lingshu-7B, our method achieves 108.27% of the baseline average performance. A similar trend is observed on HuatuoGPT-Vision-7B, where our approach yields a 104.16% performance compared with the baseline. Notably, competing pruning methods such as VisionZip, VScan, FastV, and PDrop generally maintain or slightly degrade performance compared to the baseline, reflecting the inherent difficulty of preserving task-critical visual evidence under heuristic token selection. We attribute our gains to the proposed dual-stream RL mechanism together with the cross-feedback sequential optimization. GTP acts as an inference-time heuristic. In contrast, DS-RL internalizes this grounding-aware evidence selection ability into the model by casting visual token pruning as a policy-guided learning process. Therefore, the model can actively focus on compact yet semantically sufficient evidence and reason effectively over sparse visual tokens.

In addition to performance improvement, we compare inference efficiency with strong visual token pruning baselines.

In Table 4, our method achieves an around ×4.5 speedup on Lingshu-7B and an around ×3.4 speedup on Lingshu-32B compared to competing methods. We attribute this efficiency advantage to the fact that our approach is still compatible with the vLLM inference framework. Meanwhile, existing visual token pruning methods heavily rely on visual similarity computations from visual encoder or attention-based statistics extracted from the LLM decoder, which prevents efficient inference acceleration and also limits their applicability to end-to-end training. Therefore, even without vLLM acceleration, our method still completes inference in 31 minutes on Lingshu-7B and 81 minutes on Lingshu-32B, remaining faster than competing methods. By formulating VTP as a policy-guided decision process, our method remains fully compatible with efficient inference engines as well as RL-based training optimization. This design demonstrates strong potential in jointly improving training scalability, domain-specific task performance, and inference efficiency.

### 4.4. Comparison with Medical RL Methods

We report the main results compared with existing medical RL methods in Table 3. Overall, ViToS consistently outperforms strong RL medical baselines across most benchmarks and achieves the best average performance, indicating its strong generalization across diverse medical scenarios. ViTAR (Chen et al., 2025a) is designed to focus on multi-round reasoning and mimic expert decision-making behaviors through iterative interactions, and is trained on the same dataset as our method. In contrast, ViToS leverages token-sparse reasoning via grounding-aware VTP, which

*Table 2.* Performance comparison of different visual token pruning strategies across multiple medical VQA benchmarks. Our approach reformulates visual token pruning as a policy-controlled evidence selection process via a dual-stream RL and cross-feedback sequential optimization, leading to consistent accuracy improvements over both the baseline models and existing pruning methods.

| | PMC-VQA | PathVQA | SLAKE | VQA-RAD | Omni. | MMMU-Med | MedX. | Average | Percentage |
|---|---|---|---|---|---|---|---|---|---|
| *Lingshu-7B* | | | | | | | | | |
| Baseline | 60.20 | 71.98 | 81.25 | 64.14 | 81.34 | 63.33 | 23.50 | 63.68 | 100.00% |
| VisionZip | 54.63 | 73.26 | 81.01 | 64.14 | 81.20 | 62.76 | 23.47 | 62.92 | 98.82% |
| VScan | 53.90 | 73.26 | 81.97 | 64.94 | 81.27 | 64.83 | 23.09 | 63.32 | 99.44% |
| FastV | 51.60 | 71.48 | 80.53 | 61.75 | 77.58 | 56.55 | 22.39 | 60.27 | 94.65% |
| PDrop | 54.00 | 73.17 | 82.69 | 61.75 | 81.06 | 66.67 | 23.51 | 63.26 | 99.35% |
| Ours | 62.40 | 77.19 | 84.62 | 71.31 | 83.30 | 78.00 | 25.80 | 68.95 | 108.27% |
| *HuatuoGPT-Vision-7B* | | | | | | | | | |
| Baseline | 55.05 | 65.76 | 70.91 | 66.53 | 71.72 | 45.33 | 22.35 | 56.81 | 100.00% |
| VisionZip | 55.60 | 66.00 | 71.15 | 67.33 | 74.08 | 44.00 | 17.50 | 56.52 | 99.50% |
| VScan | 55.40 | 65.68 | 71.88 | 67.73 | 73.86 | 46.00 | 17.30 | 56.84 | 100.05% |
| FastV | 50.05 | 64.93 | 70.19 | 63.75 | 72.52 | 44.83 | 21.06 | 55.33 | 97.40% |
| PDrop | 49.20 | 64.04 | 69.23 | 66.53 | 73.53 | 44.00 | 21.95 | 55.50 | 97.69% |
| Ours | 52.40 | 63.80 | 78.61 | 70.52 | 74.96 | 50.00 | 23.90 | 59.17 | 104.16% |
| *Lingshu-32B* | | | | | | | | | |
| Baseline | 61.70 | 77.28 | 85.82 | 70.52 | 79.52 | 62.00 | 28.40 | 66.46 | 100.00% |
| VisionZip | 62.90 | 77.75 | 90.62 | 63.75 | 78.34 | 58.67 | 22.45 | 64.93 | 97.69% |
| VScan | 62.70 | 77.36 | 90.14 | 66.93 | 78.65 | 60.00 | 22.20 | 65.43 | 98.44% |
| FastV | 61.75 | 77.16 | 87.50 | 63.35 | 77.94 | 59.31 | 28.21 | 65.03 | 97.85% |
| PDrop | 61.95 | 77.31 | 88.94 | 64.14 | 77.71 | 58.67 | 28.45 | 65.31 | 98.27% |
| Ours | 63.05 | 85.37 | 90.62 | 72.51 | 83.45 | 66.67 | 29.80 | 70.21 | 105.64% |

*Table 3.* Main results on seven medical multimodal benchmarks, comparing ViToS with strong medical RL baselines. ViToS achieves improvements over the competing RL methods. These results show that reformulating VTP as policy-guided evidence selection enables the model to focus on relevant visual evidence and improves token-sparse medical multimodal reasoning.

| | PMC-VQA | PathVQA | SLAKE | VQA-RAD | Omni. | MMMU-Med | MedX. | Average |
|---|---|---|---|---|---|---|---|---|
| Med-R1 | 45.8 | 53.3 | 55.1 | 55.9 | - | 32.7 | 20.3 | - |
| MedVLM-R1 | 44.8 | 55.2 | 65.9 | 61.4 | - | 35.5 | 21.2 | - |
| MedCCO | 53.2 | - | - | - | 65.8 | 59.3 | 23.2 | - |
| MedVLThinker | 51.6 | 66.0 | 64.1 | 64.9 | 67.8 | 45.5 | 23.4 | 54.8 |
| ViTAR | 57.2 | 67.0 | 80.8 | 70.1 | 74.2 | 72.0 | 26.9 | 64.0 |
| **ViToS (Ours)** | 62.4 | 77.2 | 84.6 | 71.3 | 83.3 | 78.0 | 25.8 | 68.9 |

substantially improves training efficiency. As a result, ViToS (8 hours) requires approximately 1/3 of the training time compared to ViTAR (23 hours), while achieving better performance. This comparison highlights that token-sparse reasoning not only reduces computational cost but also leads to more effective learning for medical reasoning.

*Table 4.* Inference efficiency Comparison with existing visual token pruning methods (time: minutes).

| Scales | VisionZip | VScan | FastV | PDrop | ViToS |
|---|---|---|---|---|---|
| 7B | 50 | 54 | 52 | 47 | **11** |
| 32B | 172 | 184 | 171 | 202 | **53** |

## 4.5. Impact of Visual Redundancy Across Modalities

To examine the impact of visual redundancy across different medical imaging modalities, we calculate the average pairwise distance among visual tokens in the image as the token distance, and we analyze the performance gains with the average pairwise distance. As reported in Appendix Figure 5, the token distance serves as a quantitative indicator of visual redundancy, where smaller values correspond to higher similarity among tokens. Across all evaluated modalities, visual token pruning consistently yields performance improvements over the baseline, demonstrating its general effectiveness. To better isolate the effect of visual redundancy on the effectiveness of visual token pruning, we conduct a controlled analysis by comparing modalities with similar accuracy. Notably, although X-ray and fundus

*Table 5.* Ablation study on cross-feedback and sequential optimization. Only localization branch (Branch-L) optimization yields moderate gains, adding cross-feedback (CF) rewards improves performance further, and full sequential optimization combining localization and token-sparse reasoning (Branch-L&S) achieves the best results. Epochs used for each phase are listed for reference.

| | PMC-VQA | PathVQA | SLAKE | VQA-RAD | Omni. | MMMU-Med | MedX. | Average |
|---|---|---|---|---|---|---|---|---|
| Baseline | 60.20 | 71.98 | 81.25 | 64.14 | 81.34 | 63.33 | 23.50 | 63.68 |
| Branch-L w/o CF (2 epoch) | 61.00 | 74.99 | 81.25 | 67.73 | 82.23 | 68.00 | 25.35 | 65.79 |
| Branch-L (2 epoch) | 60.75 | 75.97 | 82.69 | 69.32 | 83.11 | 77.33 | 25.30 | 67.78 |
| Branch-L (3 epoch) | 60.95 | 76.86 | 81.25 | 70.92 | 83.30 | 77.33 | 25.25 | 67.98 |
| Branch-L&S (3 epoch) | 62.40 | 77.19 | 84.62 | 71.31 | 83.30 | 78.00 | 25.80 | 68.95 |

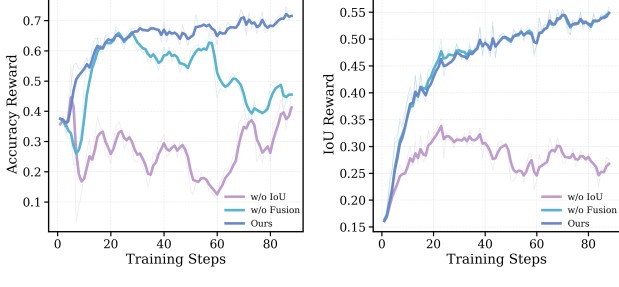

*(a)* Accuracy reward trends.     *(b)* IoU reward trends.

*Figure 4.* Comparison of accuracy and IoU reward trends under settings without the IoU signal or without token fusion.

imaging achieve comparable baseline performance, fundus images exhibit substantially larger performance gains after token pruning, which aligns well with their higher degree of visual token redundancy. A similar trend can be observed when comparing dermoscopy with ultrasound, as well as MR with microscopy. This key finding suggests that visual token pruning is particularly advantageous for medical imaging modalities with highly redundant visual patterns.

### 4.6. Ablation Study on Cross-Feedback and Sequential Optimization

Table 5 presents an ablation study analyzing the impact of cross-feedback and sequential optimization on model performance. When only the localization branch is optimized for two epochs without cross-feedback (Branch-L w/o CF), the model already achieves a noticeable improvement over the baseline, indicating that RL-based localization optimization is beneficial. When cross-feedback from the token-sparse reasoning branch is incorporated (Branch-L), the model achieves a substantial additional performance boost, particularly on benchmarks such as VQA-RAD and MMMU-Med. This result highlights the critical role of downstream reasoning as a cross-feedback signal, guiding the localization policy to attend to visual regions that preserve task-critical evidence and support reliable inference. Further extending localization branch optimization to three epochs yields only marginal gains, suggesting that localization learning quickly saturates and is no longer the primary

performance bottleneck. In contrast, enabling sequential optimization combining localization and token-sparse reasoning (Branch-L&S) results in the best overall performance across all datasets. This confirms that sequential optimization with cross-feedback is crucial for stabilizing training and enabling effective reasoning under pruned visual representations. Overall, these ablation results validate the necessity of both cross-feedback and sequential optimization, revealing their complementary roles in learning token-sparse reasoning within a unified dual-stream RL framework.

### 4.7. Role of Token Fusion in Dual-Branch Training

We investigate the effect of removing the token fusion mechanism in the token-sparse reasoning branch. Without token fusion, visual tokens outside the predicted bounding box are directly discarded. Training accuracy exhibits fluctuations between epochs 60 and 80 when token fusion is disabled (Figure 4a). This instability arises because inaccurate bounding box predictions may remove task-critical visual tokens, leaving the reasoning branch with insufficient or misleading evidence. By contrast, the token fusion softly aggregates background information into retained tokens, preserving global cues and ensuring stable learning even under imperfect localization. Consistent results are also observed in the final test performance in the Appendix Figure 9.

### 4.8. Impact of IoU Reward

In the localization branch, we employ an IoU-based reward to provide explicit spatial supervision. To analyze the role of this signal, we conduct ablation experiments by removing the IoU reward and training the model using only accuracy- or format-based rewards. Experimental results show that without the IoU supervision signal, the model struggles to acquire reliable spatial alignment with low IoU reward in Figure 4b. The localization branch fails to consistently select task-relevant visual tokens, which in turn deprives the reasoning branch of meaningful visual evidence, leading to pronounced training instability in Figure 4a. This instability in localization further translates into inferior testing performance, as detailed in the the Appendix Figure 9. These findings demonstrate that an explicit localization

stream with IoU reward is essential in our dual-stream RL framework. See more results in Appendix Section A.4.

## 5. Conclusion

In this paper, we underscore that focusing on grounded tokens via VTP can effectively mitigate visual distraction and boost reasoning ability. We recognize that grounding-aware VTP should be treated as a policy-guided process to optimize token-sparse medical multimodal reasoning. To respond, we propose ViToS, a novel dual-stream RL framework that learns token pruning and token-sparse reasoning. In a unified model, ViToS learns grounding and token-sparse reasoning via coupled policy learning with cross-feedback sequential optimization. Extensive experiments across diverse medical benchmarks and model scales demonstrate consistent performance gains alongside reductions in visual tokens and inference cost. These findings establish that principled visual token selection is critical for achieving efficient and strong medical multimodal reasoning.

## Limitations and Future Work

Scaling to extreme-resolution medical images can introduce a larger localization action space as a challenge for policy learning. In this study, we follow the standard set-up from related works by adopting standard-resolution medical image, where ViToS can be extensively developed and evaluated. Our study is not specifically designed for such extreme-resolution settings. Extending our framework to handle extreme-resolution inputs (e.g., via hierarchical strategies for localization) is an important direction for future work.

## Acknowledgment

This work was supported by Shanghai Artificial Intelligence Laboratory.

## Impact Statement

ViToS is a dual-stream reinforcement learning framework, aiming to improve medical multimodal reasoning. The grounding-aware visual token pruning and sparse-token reasoning within ViToS lay a basis for applicable medical co-pilot, offering interpretable evidence for medical decision making and superior inference speedup, which will eventually benefit the healthcare industry. This work also solves a general coupled policy learning problem via the cross-feedback sequential optimization, which is in principle instructive for other reinforcement learning research. All benchmarks and baseline models are publicly available and there are no ethical issues.

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

## A. Additional Experiments

### A.1. The Order of Training in Sequential Optimization

We adopt sequential optimization because token-sparse reasoning is inherently more challenging and strongly depends on reliable visual localization. In our default setting, we first optimize the localization branch to establish stable visual grounding, and then shift the training focus to the token-sparse reasoning branch. This training order allows the model to leverage reliable localization results for effective token selection and token-sparse reasoning.

To examine the necessity of this sequential order, we conduct an ablation study in which the model is trained by directly optimizing the sparse reasoning branch without pretraining the localization branch. As shown in Figure 6, this order leads to training failure and the model fails to converge. Without a well-learned localization policy, sparse token selection becomes highly unstable, resulting in noisy reward signals and ineffective policy updates. These results demonstrate that token-sparse reasoning cannot be learned in isolation and requires progressively established visual grounding. Therefore, sequential optimization with an appropriate training order is essential for stable training.

### A.2. Explosion of $\lambda$ in Rewards

We assign an equal weight $\lambda$ to each task-level reward (i.e., IoU and accuracy), while the format reward is weighted by $(1 - \lambda)$ and scaled according to the number of task-level rewards in each branch. We explore three values of $\lambda$, 0.8, 0.9 (default), and 0.95. From the training accuracy curves, we observe that $\lambda = 0.9$ yields the most stable optimization behavior in Figure 7. In contrast, when $\lambda = 0.95$, noticeable training instability emerges between epochs 25 and 40, while $\lambda = 0.8$ exhibits fluctuations during the early training phase (within the first 10 epochs).

### A.3. Role of Token Fusion in GTP

We analyze the necessity of incorporating token fusion into the grounding-aware visual token pruning (GTP) framework. Specifically, we compare GTP with pruning only (without token fusion) against the baseline. In Figure 8, without token fusion, GTP yields consistent improvements on VQA benchmarks, achieving average accuracy gains of 1.1% and 3.6% on the 7B and 32B Lingshu models, respectively. This suggests that removing redundant visual tokens can partially suppress noise and improve discriminative performance. However, this pruning-only strategy exhibits notable instability across datasets. When foreground localization is inaccurate, visually informative tokens may be mistakenly pruned, leading to severe performance degradation. This issue is particularly pronounced in tasks requiring multi-region reasoning (e.g., PMC-VQA) or spatial reasoning (e.g., SLAKE), where pruning disrupts global context and inter-region dependencies. As a result, performance drops by 5.4% on PMC-VQA and 4.3% on SLAKE for Lingshu-7B. These results indicate that merely discarding redundant visual information is insufficient for robust medical visual reasoning. To address this limitation, we introduce a token fusion that integrates information from non-critical regions into foreground visual tokens. This design preserves foreground semantic saliency while maintaining global contextual cues. Token fusion effectively mitigates the

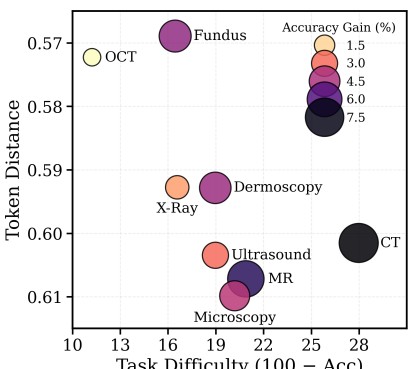

*Figure 5.* Impact of visual redundancy across eight medical modalities.

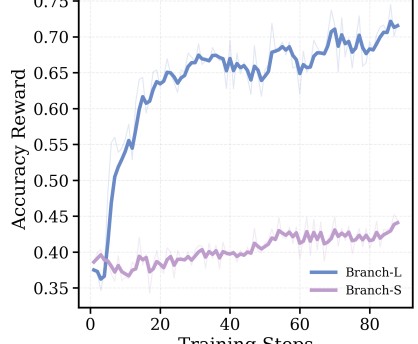

*Figure 6.* Ablation study on the training order. Branch-L and Branch-S correspond to training the localization branch or the token-sparse reasoning branch first, respectively.

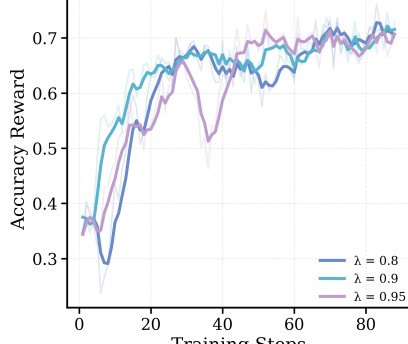

*Figure 7.* Accuracy reward trends under different values of the weighting parameter $\lambda$: 0.8, 0.9 (default), and 0.95.

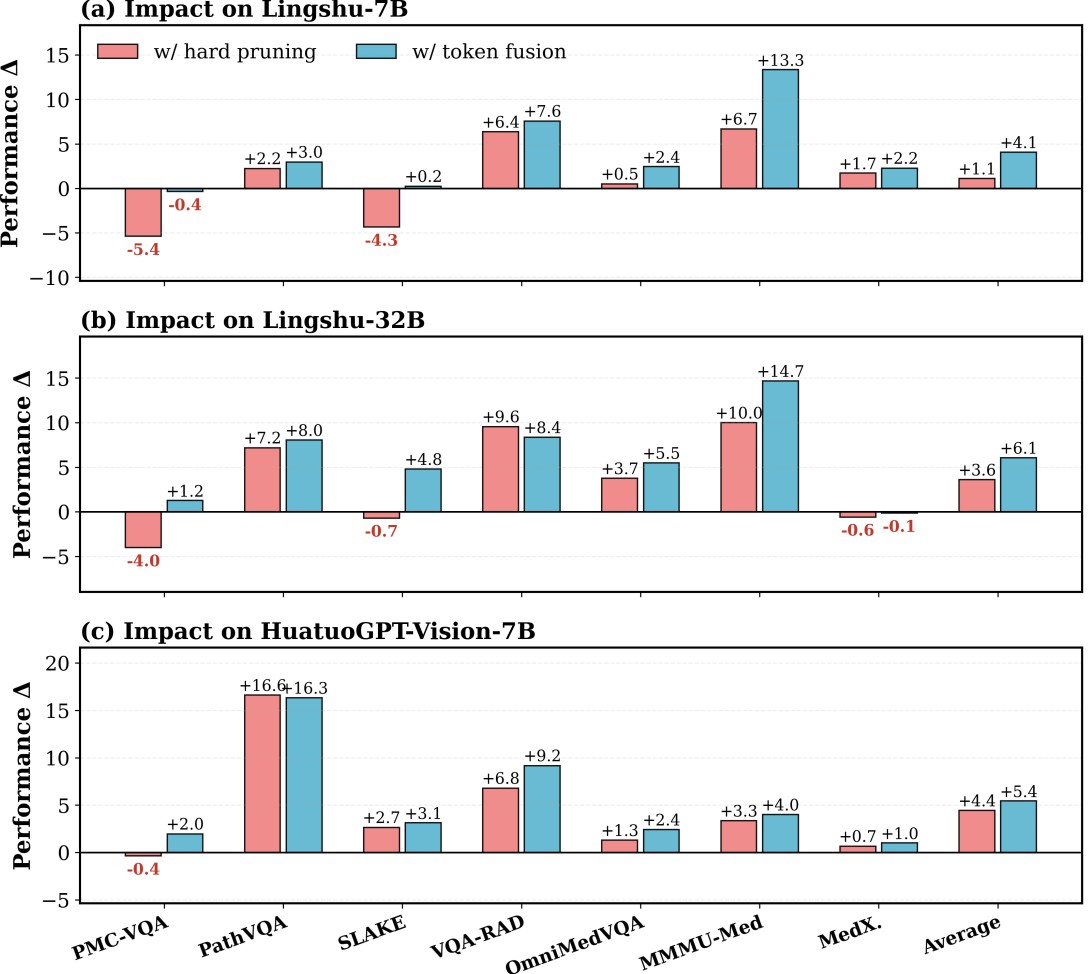

*Figure 8.* Ablation study about token fusion in the grounding-aware visual token pruning.

reasoning instability caused by hard pruning, demonstrating that it is a necessary component of the GTP framework.

### A.4. Role of Token Fusion and IoU Reward in DS-RL Training

In addition to training dynamics we discussed in Section 4.7 and 4.8, we report test-time performance comparisons across different ablation settings in Figure 9. As shown by the results, our full model consistently outperforms both variants without IoU reward or token fusion during DS-RL training across evaluation benchmarks. Specifically, removing the IoU reward leads to notable performance degradation, indicating that explicit spatial supervision is critical for learning reliable localization policies. Similarly, disabling token fusion also results in reduced performance compared to the full model. While the degradation is generally less severe than removing IoU supervision, the model without token fusion shows weaker robustness, confirming that softly preserving background context helps mitigate localization errors.

Overall, these results are consistent with the training-time analysis presented in the main Section 4.7 and 4.8. They demonstrate that both IoU reward and token fusion play complementary roles. The former establishes accurate visual grounding, while the latter stabilizes token-sparse reasoning under imperfect localization. Together, they are essential for achieving strong and robust performance in our DS-RL framework.

### A.5. Analysis with Supervised Fine-Tuning

We adopt RL rather than supervised fine-tuning (SFT) as the training paradigm for three key reasons. First, existing medical SFT data are not suitable for training an explicit localization policy. Most SFT-oriented medical instruction datasets, such as

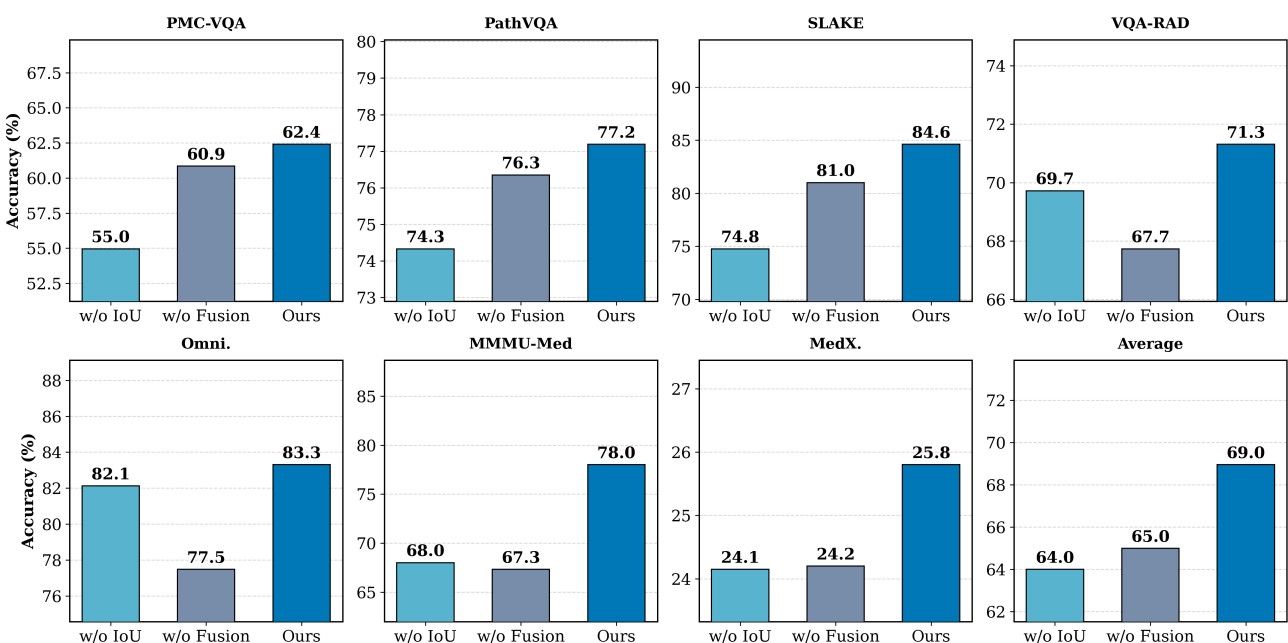

*Figure 9.* Test-time performance comparison under different ablation settings. We compare the our full model with variants without IoU reward or without token fusion across multiple medical benchmarks.

data of LLaVA-Med (Li et al., 2023) and HuatuoGPT-Vision (Chen et al., 2024a), do not provide explicit bounding box supervision. As a result, they cannot support learning a dedicated localization branch that outputs spatial coordinate for downstream reasoning. Second, SFT is inherently ill-suited for optimizing discrete spatial decisions such as coordinate prediction. SFT relies on autoregressive token-level likelihood maximization. This formulation makes it difficult to directly optimize spatial objectives such as IoU. RL naturally addresses this limitation by allowing IoU reward functions to be defined directly on geometric outcomes, enabling effective optimization of localization accuracy. Third, SFT is a passive learning paradigm that depends on mimicking predefined reasoning patterns from annotated datasets. RL plays a critical role in eliciting reasoning behaviors through active exploration, without relying on large-scale curated data. In our framework, the reasoning process emerges through interaction with sparse visual evidence and reward-driven policy updates.

In addition to the conceptual limitations discussed above, we empirically evaluate the feasibility of SFT under the same training data and model architecture. Specifically, we train the model to generate localization outputs and reasoning responses jointly using `LLaMA-Factory` (Zheng et al., 2024) training framework. However, we observe that SFT fails to converge. In the localization branch, the model is unable to consistently produce coordinates in the required fixed format, which prevents the outputs from being reliably consumed by the subsequent sparse reasoning branch. Moreover, the autoregressive language modeling loss does not decrease during training, indicating optimization failure.

### A.6. Comparison with Cropping Baselines

We conduct experiments comparing our method with both image-level cropping and token-level cropping baselines. As shown in results, naive image cropping and token cropping only achieve limited improvements over the base model, while our ViToS achieves remarkable performance gain. Notably, cropping methods exhibit inconsistent behavior across datasets with degrading performance (e.g., PMC-VQA, SLAKE), whereas our method consistently improves results. We further compare with general-domain methods that rely on direct image cropping in reasoning, such as DeepEyes (Zheng et al., 2025) and PixelReasoner (Su et al., 2025). We observe that these general VLMs struggle to perform well on medical images, highlighting the importance of domain-specific GTP rather than coarse image cropping. To illustrate, this difference stems from the fact that cropping performs hard spatial truncation and may discard clinically relevant contextual information. In contrast, our method conducts grounding-aware, token-level evidence selection and preserves contextual cues via token fusion, leading to more stable and effective reasoning.

*Table 6.* Comparison with image-level cropping, token-level cropping, and cropping-based reasoning methods.

| | PMC-VQA | PathVQA | SLAKE | VQA-RAD | Omni. | MMMU-Med | MedX. | Average |
|---|---|---|---|---|---|---|---|---|
| Base | 60.2 | 72.0 | 81.3 | 64.1 | 81.3 | 63.3 | 23.5 | 63.7 |
| Image Crop | 53.6 | 74.8 | 74.0 | 70.5 | 80.4 | 70.7 | 25.1 | 64.2 |
| Token Crop | 54.8 | 74.2 | 76.9 | 70.5 | 81.8 | 70.0 | 25.2 | 64.8 |
| DeepEyes | - | 52.9 | 68.2 | 65.9 | 64.8 | 57.8 | 23.6 | - |
| PixelReasoner | - | 52.6 | 67.3 | 66.0 | 64.9 | 58.0 | 23.5 | - |
| ViToS (Ours) | 62.4 | 77.2 | 84.6 | 71.3 | 83.3 | 78.0 | 25.8 | 68.9 |

# B. Pseudo-Code for Grounding-Aware Visual Token Pruning

---

**Algorithm 1** Pseudo-Code for Grounding-Aware Visual Token Pruning

---

**Require:** Visual tokens $\mathbf{E} \in \mathbb{R}^{N \times D}$, Predicted bounding box $b_{\text{pred}}^{(g)}$

**Ensure:** Sparse visual representation $\hat{\mathbf{E}}_{\text{kept}}^{(g)}$

1: **Partitioning:**
2: Identify tokens inside $b_{\text{pred}}^{(g)}$
3: Divide visual tokens $\mathbf{E}$ into foreground tokens $\mathbf{E}_{\text{kept}}^{(g)}$ and background tokens $\mathbf{E}_{\text{others}}^{(g)}$
4: **if** $|\mathbf{E}_{\text{others}}^{(g)}| = 0$ **then**
5: $\quad \mathbf{E}_{\text{kept}}^{(g)}$
6: **end if**
7: **Normalization & Similarity Computation:**
8: $\bar{\mathbf{E}}_{\text{kept}} \leftarrow \text{Normalize}(\mathbf{E}_{\text{kept}}^{(g)})$
9: $\bar{\mathbf{E}}_{\text{others}} \leftarrow \text{Normalize}(\mathbf{E}_{\text{others}}^{(g)})$
10: Compute similarity matrix $\mathbf{S} = \bar{\mathbf{E}}_{\text{others}}(\bar{\mathbf{E}}_{\text{kept}})^{\top}$
11: **Token Merging:**
12: **for** each background token $e_j \in \mathbf{E}_{\text{others}}^{(g)}$ **do**
13: $\quad$ Find nearest foreground index: $k_j \leftarrow \arg\max_i \mathbf{S}[j, i]$
14: $\quad$ Assign $e_j$ to set $\mathcal{C}_{k_j}$ associated with foreground token $i = k_j$
15: **end for**
16: **for** each foreground token $e_i \in \mathbf{E}_{\text{kept}}^{(g)}$ **do**
17: $\quad$ Calculate aggregated feature:
18: $\quad \hat{e}_i \leftarrow \frac{e_i + \sum_{e_j \in \mathcal{C}_i} e_j}{1 + |\mathcal{C}_i|}$
19: **end for**
20: Construct $\hat{\mathbf{E}}_{\text{kept}}^{(g)}$ by stacking all $\hat{e}_i$

---

# C. Supplementary Experiment Details

## C.1. Details of Section "Main Results"

We evaluate the proposed methods on two representative medical VLMs, namely Lingshu[2,3] and HuatuoGPT-Vision[4]. Lingshu covers two model scales: 7B and 32B. All baselines are based on officially released checkpoints on HuggingFace and are evaluated under a unified prompt (see Figures 10 and 11). Inference is performed with deterministic greedy decoding (temperature = 0) and the thinking process is disabled to reduce inference time. All models are accelerated using vLLM with 0.9.2 version to ensure fair comparison and consistency.

---

[2] https://huggingface.co/lingshu-medical-mllm/Lingshu-7B
[3] https://huggingface.co/lingshu-medical-mllm/Lingshu-32B
[4] https://huggingface.co/FreedomIntelligence/HuatuoGPT-Vision-7B-Qwen2.5VL

**C.2. Details of Section "Comparison with Previous Pruning Methods"**

In our methods, the average token retention is 77%. For baseline pruning methods that require a fixed retention ratio, including VisionZip, VScan, FastV, and PDrop, we uniformly set the retention rate to 80% to ensure a fair comparison. Specifically, for VisionZip, the dominant token selection is set to 80%; for VScan, token pruning is applied only during the complementary global and local stages of stage I, and no pruning is performed in the LLM decoder; for FastV, pruning is applied before the first LLM interaction; and for PDrop, pruning is performed in the middle 15 layers (defaults settings) of the model. VisionZip and VScan are implemented using their official GitHub code[5,6]. Note that FastV and PDrop officially support only the LLaVA architecture and do not natively support Qwen-VL, thus we manually adapted these methods to the Qwen-VL architecture for evaluation.

## D. Training Details

For training, we adopt the `veRL` framework combined with the GRPO advantage estimator. Due to the large number of rollouts requiring operations on visual tokens, we perform a pre-processing step to generate and save the visual tokens prior to training. During training, these visual encoder of VLMs are frozen, which significantly reduces training time. To accommodate models of different scales, we use distinct key training parameters for the 7B and 32B models. Specifically, the maximum prompt length is set to 8,192 for the 7B model and reduced to 2,048 for the 32B model; the rollout group size is 8 and 5, respectively; and the rollout batch size per step is maintained at 384. Micro-batch configurations are also adjusted: the actor uses a micro-batch of 8 for gradient updates in the 7B model and 1 in the 32B model, while the experience sampling micro-batch is 16 for 7B and 2 for 32B model. The tensor parallelism scale on GPU is increased from 2 to 8. The total number of training epochs is 3, and the global batch size is 96, with optimizer state offloading enabled for both models. The optimization strategy is `adamw` for the 7B model and `adamw-bf16` for the 32B model.

## E. Training Data

We adpot ViTAR (Chen et al., 2025a) training dataset for RL training. Following the same data construction workflow as ViTAR, we utilize DeepSeek-V3-671B (Liu et al., 2024) to regenerate and re-validate all VQA samples, ensuring data quality. The dataset is constructed based on 19 carefully curated object detection datasets and 69 task-specific templates, resulting in 18,222 VQA instances with corresponding bounding-box annotations. The training data cover a wide range of medical imaging modalities. MRI constitutes the majority and accounts for 45% of total samples while followed by ultrasound (15%), CT (12%) and X-ray (11%). The remaining portion comprises OCT (8%), pathology (6%) and electrocardiogram (3%). Overall, the dataset exhibits substantial diversity, encompassing multiple anatomical regions and a broad spectrum of disease categories.

## F. Visual Token Pruning Methods

VisionZip (Yang et al., 2025b) is a text-agnostic method for reducing visual token redundancy. It utilizes attention map of the `[CLS]` token and calculates the average self-attention scores to select the dominant tokens. A token merging strategy aggregates similar tokens to preserve important details.

VScan (Zhang et al., 2025a) is a two-stage token pruning approach which can be divided into pre-LLM stage and mid-LLM stage. At the pre-LLM stage, it extract locally important tokens from shallow layers and globally critical tokens from output layers. The other tokens will be merged into selected tokens. At the mid-LLM stage, VScan utilizes attention of textual tokens to prune visual tokens at the intermediate layers of LLM.

FastV (Chen et al., 2024b) allow higher-resolution image processing without slowing down inference. It show that not all tokens are equally valuable in visual attention, leading to smarter token selection and pooling strategies. It selects an early layer of LLM and calculates average self-attention scores of each visual token. In addition, some tokens of deep layers will be discarded directly.

PDrop (Xing et al., 2024) divides the layers of the LLM into several stages. It retains a large part of visual tokens in shallow layers and gradually reduces the retention ratio in deep layers. Whether a visual token is kept is determined by the attention

---

[5] https://github.com/JIA-Lab-research/VisionZip
[6] https://github.com/Tencent/SelfEvolvingAgent/tree/main/VScan

scores of the instruction token.

## G. Prompts

---
**Prompt for Localization Branch**

You need to analyze the given medical image. Provide your detailed, step-by-step reasoning based on the image. Then, produce detailed annotations focus on the key regions, ending with the final answer.
Instructions:
1. Perform a step-by-step reasoning based on the image analysis. Enclose this reasoning process between: `<think>` Your reasoning process here `</think>`
2. Focus on the key regions based on question, produce a bounding box in the format: `<bbox>` [[x1, y1, x2, y2], ...] `</bbox>`
- 'x1, y1' = normalized coordinates of the top-left corner (values in range 0.0 - 1.0)
- 'x2, y2' = normalized coordinates of the bottom-right corner (values in range 0.0 - 1.0)
3. Conclude by providing the final option letter to the question in: `\boxed{FINAL ANSWER}`
Output format example:
`<think>` Your reasoning process here `</think>` `<bbox>` [[x1, y1, x2, y2], ...] `</bbox>` `\boxed{FINAL ANSWER}`

---

*Figure 10.* Prompt for localization branch.

---
**Prompt for Token-Sparse Reasoning Branch**

Determine the answer to the question. First provide an internal step-by-step reasoning within `<think>` `</think>` then provide a option letter in `\boxed{FINAL ANSWER}`

---

*Figure 11.* Prompt for token-sparse reasoning branch.

## H. Case Study

In Figure 12, the model with sparse grounded token focuses on diagnostically critical visual evidence, including a well-circumscribed oval mass with coarse "popcorn" calcifications, and correctly identifies the lesion as a benign fibroadenoma. In contrast, the model with full token overestimates malignancy, leading to an incorrect diagnosis.

In Figure 13, the model with sparse grounded token precisely localizes the contrast-enhanced tubular structure, identifying it as the azygous vein in agreement with expert annotation. By grounding its reasoning in exact spatial and anatomical cues, it avoids confusing vascular anatomy with surgical artifacts. Conversely, the full-token model over-relies on diffuse contextual signals, incorrectly predicting a Blalock–Taussig shunt.

In Figure 14, the model with sparse grounded token accurately attends to multi-layered retinal hemorrhages and exudative changes, correctly identifying shaken-baby syndrome in line with human expert assessment. By concentrating on key pathological features rather than overall retinal appearance, it avoids misdiagnoses such as retinoblastoma or retinopathy of prematurity. The full-token model, however, overemphasizes spurious patterns and misclassifies the condition as retinoblastoma.

In Figure 15, the model with sparse grounded token selectively focuses on salient radiographic signs, including extensive periarticular soft tissue calcifications and joint deformities, correctly diagnosing scleroderma consistent with expert evaluation. By attending to clinically relevant cues, it avoids being misled by surrounding bone or soft tissue. In contrast, the full-token model considers multiple less-relevant features, producing ambiguous and conflicting predictions.

These examples illustrate that sparse grounding suppresses visual noise and yields predictions more consistent with the human expert assessment. Beyond improving diagnostic accuracy, the model with sparse token also enhances the

Question: What is the MOST likely etiology of the mass indicated by the arrow? A) Phyllodes tumor B) Invasive ductal carcinoma with DCIS C) Metastasis from a primary D) Intracystic papillary carcinoma **E) Fibroadenoma**

*Model w/ full tokens*

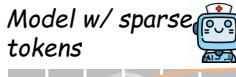

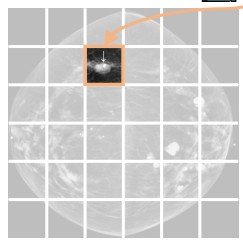

<think>The image provided is a mammogram showing a mass indicated by an arrow. To determine the etiology of the mass, we need to analyze its characteristics. The mass appears to have **irregular margins and a heterogeneous density**, which are concerning features often **associated with malignancy**. Given the options, we can rule out fibroadenoma (E) as it typically presents as a well-circumscribed, smooth mass. Phyllodes tumor (A) can also present with irregular margins but is less common. Metastasis from a non-breast primary (C) is possible but less likely given the typical presentation of primary breast lesions. Intracystic papillary carcinoma (D) usually presents as a cystic lesion, which does not match the appearance here. Invasive ductal carcinoma with DCIS (B) is the most likely etiology due to the **irregular margins and heterogeneous density**. Therefore, the most probable answer is B.</think>
<bbox> [[0.26, 0.16, 0.42, 0.27]] </bbox> <answer>**B**</answer>

*Model w/ sparse tokens*

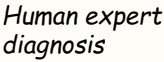

<think>To determine the etiology of the mass indicated by the arrow, we must analyze the mammographic features and correlate them with clinical knowledge. The mass appears **well-defined and oval-shaped**, which is characteristic of **benign lesions such as fibroadenomas**. Fibroadenomas often present as smooth, circumscribed masses on mammography. In contrast, malignant lesions like invasive ductal carcinoma or DCIS typically exhibit irregular margins, spiculations, or microcalcifications. Phyllodes tumors, while also benign, tend to have a more heterogeneous appearance and may show rapid growth. Metastasis from a non-breast primary is less likely given the typical presentation of metastatic lesions in the breast, which often appear as multiple masses. Intracystic papillary carcinoma would typically present with cystic features, which are not evident in this image. Based on the **smooth, well-defined appearance of the mass**, the most likely etiology is a fibroadenoma.</think> <answer>**E**</answer> ✓

*Human expert diagnosis*

The arrow points to a **well-defined, oval, high-density mass** with coarse 'popcorn' calcifications, consistent with **a benign involuting fibroadenoma**.

*Figure 12.* Comparison between full token and sparse grounded token reasoning for mass diagnosis.

interpretability of visual reasoning, pointing toward a paradigm where AI system can support clinical decision-making with greater precision and trust.

Question: "Which anatomical structure is demonstrated with contrast enhancement? A) Interlobar mesothelioma **B) Azygous vein** C) Blalock-Taussig shunt D) Pulmonary embolism E) Cervical rib

*Model w/ full tokens* 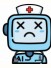

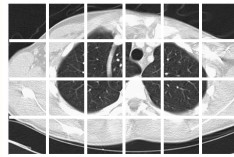

<think>The image provided is a cross-sectional CT scan of the chest, showing the lungs and surrounding structures. The question asks about the anatomical structure demonstrated with contrast enhancement. To determine the correct answer, we need to analyze the image for any enhanced structures and correlate them with the given options. Upon examining the image, there is a tubular structure with contrast enhancement located near the mediastinum, which is consistent with **the appearance of a vascular shunt**. Similarly, the appearance does not match interlobar mesothelioma, which would present as a pleural-based lesion, nor does it resemble a pulmonary embolism. The structure also does not correspond to a cervical rib, which would be seen as an osseous structure. Therefore, the most likely answer is the Blalock-Taussig shunt</think> <bbox> [[0.39, 0.3, 0.68, 0.67]] </bbox> <answer>*C*</answer>

*Model w/ sparse tokens* 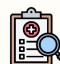

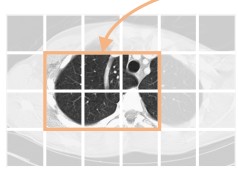

<think>To determine the correct answer, we need to analyze the CT image and identify the anatomical structure. The image shows a cross-sectional view of the chest, including the lungs, mediastinum, and surrounding structures. The contrast enhancement is visible in **a tubular structure** located posterior to the trachea and anterior to the vertebral column. This structure is consistent with the **azygous vein**, which is a major vein that drains blood from the chest wall and upper abdomen into **the superior vena cava**. (A) interlobar mesothelioma, is a pathological condition and not a structure that would be enhanced with contrast. (C) Blalock-Taussig shunt, is a surgical procedure and not a naturally occurring structure. (D) pulmonary embolism, refers to a blockage in the pulmonary arteries, which would not appear as a contrast-enhanced structure. (E) cervical rib, is an anatomical variant of the rib and would not be enhanced with contrast. Therefore, the correct answer is B, azygous vein.</think> <answer>**B**</answer> ✓

*Human expert diagnosis*

The contrast enhancement is visible in **a tubular structure** consistent with the **azygous vein**, which is displaced laterally to create an accessory fissure and a diagnostic azygos lobe while draining blood into **the superior vena cava.**

*Figure 13.* Comparison between full token and sparse grounded token reasoning for lungs diagnosis.

Question: What is the most likely diagnosis based on this fundus photograph of a 5-month-old female patient? A) Retinoblastoma B) Retinopathy of prematurity C) Primary infantile glaucoma **D) Shaken-baby syndrome** E) Exudative retinitis

*Model w/ full tokens*

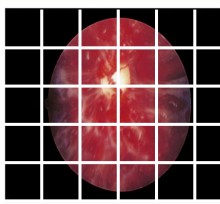

<think>The fundus photograph shows **a distinct white mass** with a central area of calcification, which is **characteristic of a retinoblastoma**. Retinoblastoma is a malignant tumor of the retina that commonly presents in young children. The appearance of the lesion in this image, aligns with the typical presentation of retinoblastoma. Other options, such as retinopathy of prematurity, primary infantile glaucoma, shaken-baby syndrome, and exudative retinitis, do not typically present with this specific appearance. Retinopathy of prematurity usually shows abnormal blood vessel growth, primary infantile glaucoma presents with corneal changes, shaken-baby syndrome often shows retinal hemorrhages, and exudative retinitis typically shows exudative changes without a central calcified mass. Therefore, the most likely diagnosis based on the image is retinoblastoma.</think> <bbox> [[0.44, 0.35, 0.65, 0.58]] </bbox> <answer>*A*</answer>

*Model w/ sparse tokens*

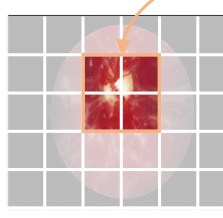

<think>The fundus photograph shows a significant amount of **retinal hemorrhage and exudative changes**, which are indicative of severe trauma or a pathological condition. Retinoblastoma (Option A) typically presents as a white mass or calcification in the retina, which is not evident here. Retinopathy of prematurity (Option B) is unlikely as it is associated with premature infants and typically presents with neovascularization, which is not seen in this image. Primary infantile glaucoma (Option C) would present with corneal changes and increased intraocular pressure, which are not visible in this fundus photograph. Shaken-baby syndrome (Option D) is a traumatic condition that can cause **severe retinal hemorrhages and exudative changes**, consistent with the findings in this image. Exudative retinitis (Option E) could present with similar findings but is less likely in this age group without a clear infectious or inflammatory cause. Based on the clinical presentation and the fundus findings, the most likely diagnosis is shaken-baby syndrome.</think> <answer>**D**</answer> ✓

*Human expert diagnosis* 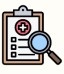

This fundus photograph shows **extensive, multi-layered retinal hemorrhages** (subretinal, intraretinal, and preretinal) extending to the periphery, which is a classic clinical hallmark of non-accidental repetitive acceleration-deceleration injury in an infant.

*Figure 14.* Our method enables accurate identification of shaken-baby syndrome by focusing on key retinal hemorrhages.

Question: Which condition is represented by this diagnosis?  **A) Scleroderma**  B) Osteosarcoma  C) Wilson's disease
D) Hypoparathyroidism E) Bullous pemphigoid

*Model w/ full tokens*

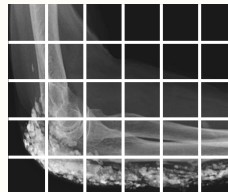

<think>The image provided is a lateral X-ray of the elbow joint. Upon close examination, there are notable features such as **subperiosteal bone resorption and calcification**, which are characteristic of **hypoparathyroidism**. These findings are consistent with the clinical presentation of hypoparathyroidism, which often leads to decreased calcium levels and secondary changes in bone structure.(A) scleroderma, typically presents with soft tissue changes and calcifications but not the specific bone changes seen here. (B) osteosarcoma, would show aggressive bone destruction and periosteal reaction, which are not evident in this image. (C) Wilson's disease, primarily affects the liver and brain and does not typically present with these bone changes. (E) bullous pemphigoid, is a skin condition and would not manifest in bone changes. Therefore, the most likely diagnosis based on the image is hypoparathyroidism.</think>
<bbox> [[0.07, 0.51, 0.61, 0.76]] </bbox> <answer>D</answer>

*Model w/ sparse tokens*

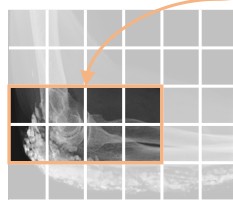

<think>To determine the correct diagnosis from the provided X-ray image, we need to analyze the radiographic features and correlate them with the clinical conditions listed. The image shows **significant soft tissue calcification and joint deformity**, which are characteristic of **a chronic connective tissue disorder**. (A) scleroderma, is a condition that involves fibrosis and calcification of the skin and underlying tissues, which aligns with the findings in the image. (B) osteosarcoma, typically presents with aggressive bone destruction, which are not evident here. (C) Wilson's disease, primarily affects the liver and brain and does not typically present. (D) hypoparathyroidism, is associated with hypocalcemia and may show subperiosteal bone resorption, but not the extensive calcification seen here. (E) bullous pemphigoid, is a skin condition characterized by blistering and does not involve calcification or joint deformity. Based on the radiographic evidence and clinical correlation, the most likely diagnosis is **scleroderma**.</think> <answer>**A**</answer> ✓

*Human expert diagnosis*

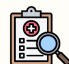

The diagnosis is **Scleroderma**, as the radiograph demonstrates **extensive periarticular soft tissue calcification** (calcinosis cutis) characteristic of systemic sclerosis, particularly the CREST variant.

*Figure 15.* Our method directs attention to soft tissue calcification and joint deformitys, enabling correct scleroderma diagnosis.

