# OpenReview forum: "Token-Sparse Medical Multimodal Reasoning via Dual-Stream Reinforcement Learning"
_ICML.cc/2026/Conference — ICML 2026 regular_

### Official Review · Reviewer_R3Qs · 2026-03-05

**Soundness:** 3
**Presentation:** 2
**Significance:** 3
**Originality:** 3
**Overall Recommendation:** 4
**Confidence:** 4

**Summary:**

The authors propose a dual-stream RL framework, ViToS, which can simultaneously optimize localization and visual pruning in an end-to-end manner. Experiments suggest that the proposed method both improves performance and alleviate computational burdens.

**Compliance With Llm Reviewing Policy:**

Affirmed.

**Final Justification:**

The authors have addressed my concerns in the design of cross-feedback (CF) reward. The explanation about the balance between localization improvement and robustness to information loss is reasonable. Simultaneously optimizing localization and visual pruning is a promising direction. Although I think SFT with high-quality CoT and grounding as a cold start would lead to better results, the framework proposed by the authors has been proved effective. As a result, I decide to raise the score to 4.

**Key Questions For Authors:**

The authors claimed in A.5 that SFT is not suitable for the task since existing medical SFT data lacks localization supervision. However, since the RL phase utilizes GT grounding boxes to calculate IoU rewards, why not leveraging this data to construct a SFT dataset? Besides, would SFT be useful as a cold start?

**Limitations:**

yes

**Strengths And Weaknesses:**

## Strengths

1. Given the failure of directly applying visual token pruning to medical MLLMs, the paper proposed a reasonable method to decouple inference process into localization and token-sparse reasoning branches.
2. Successfully train the RL framework in and end-to-end manner by sequentially optimizing grounding ability and grounding-aware VTP performance with cross-feedback rewards.
3. The experimental results show that ViToS effectively enhances performance compared with token pruning methods and medical reasoning models.

## Weaknesses

1. Unclear Rationale of Cross-Feedback Reward. In Section 3.3, the authors state that cross-feedback reward is applied by adding the reward from the frozen branch to the optimized branch training objective. This appears confused to formulate the reward for one branch with the output of another frozen branch, since the branches are decoupled in optimization. Furthermore, the ablation study in Table 5 only ablated the Cross-Feedback Reward in Branch-L (`Branch-L w/o CF` ), but missing a crucial setting to ablate this reward in Sequential optimization of both branches (`Branch-L&S w/o CF`).
2. The performance gains from Grounding-Aware Visual Token Pruning (GTP) might primarily stem from making the reasoning branch more robust to survive the information loss caused by pruning and, possibly, inaccurate bounding box predictions, rather than improving visual perception and localization. This can be implied from the need of Token Fusion, which likely compensates for unreliable localization by recovering critical information that is not correctly localized in predicted bounding boxes.

I would be willing to adjust my ratings if the concerns are addressed.

---

> ### Author Rebuttal · Authors · 2026-03-31
>
> We sincerely thank the reviewer for the insightful and constructive feedback. We are glad that you find our approach of decoupling the inference process into localization and token-sparse reasoning branches to be reasonable and well-motivated. Regarding your concerns, we provide our detailed responses below:
>
> > (Weakness 1) Unclear rationale of cross-feedback (CF) reward .
>
> We thank the reviewer for this insightful comment and clarify the design from two aspects:
>
> - Joint optimization of the two branches introduces gradient conflicts, while fully decoupled optimization can lead to policy drift, where improving one capability (e.g., localization) may degrade the other (e.g., reasoning). To address this, we introduce the cross-feedback reward. Specifically, when optimizing one branch, we incorporate the reward signal from the other (frozen) branch as an auxiliary objective. This design **encourages the shared policy model to maintain performance on both branches simultaneously**. Therefore, the cross-feedback reward serves as a regularization mechanism to stabilize coupled policy learning.
>
> - We have conducted the suggested experiment (Branch-L&S w/o CF). The results show that removing cross-feedback leads to a clear performance drop (67.05 vs. 68.95 on average), demonstrating its importance in the full dual-branch setting.
>
> |                   | **PMC-VQA** | **PathVQA** | **SLAKE** | **VQA-RAD** | **Omni.** | **MMMU-Med** | **MedX.** | **Average** |
> |-------------------|:-----------:|:-----------:|:---------:|:-----------:|:---------:|:------------:|:---------:|:-----------:|
> | Base              |    60.20    |    71.98    |   81.25   |    64.14    |   81.34   |     63.33    |   23.50   |    63.68    |
> | Branch-L&S w/o CF |    60.90    |    76.06    |   83.41   |    72.91    |   80.58   |     73.53    |   21.95   |    67.05    |
> | Branch-L&S        |    62.40    |    77.19    |   84.62   |    71.31    |   83.30   |     78.00    |   25.80   |    68.95    |
>
> > (Weakness 2) GTP gains may mainly come from robustness to information loss rather than improved perception or localization.
>
> We thank the reviewer for this insightful observation. We agree that robustness to information loss is an important aspect of our design.  As shown in our experiments (Appendix Sec. A.3 and Fig. 8), **simply discarding tokens without fusion leads to degraded performance**, indicating that both selective pruning and contextual preservation are necessary for the optimal reasoning.
>
> Importantly, our experiments also demonstrate that **localization plays a critical role**. When removing the IoU-based supervision for localization, we observe that **neither the IoU reward nor the accuracy reward shows stable improvement (Fig. 4)**, and the overall performance drops (Fig. 9). This indicates that without proper localization guidance, the model cannot effectively identify relevant visual evidence, and token fusion alone is insufficient to recover the lost information. Therefore, both rounding-aware localization and token fusion,are necessary, and neither can replace the other (see also Appendix Sec. A.4).
>
> > (Question 1) Why not construct SFT data from grounding annotations, and whether SFT can serve as a cold start.
>
> We thank the reviewer for this thoughtful question. The supervision signals in our training data consist of bounding boxes and discrete answer choices, which do not naturally align with standard chain-of-thought (CoT) SFT objectives.
>
> We construct SFT data using grounding annotations and answer labels. However, we find that **such SFT data is not well-suited for effective SFT training**. We conduct experiments to evaluate SFT as a cold start using the same data. As shown in the results, SFT consistently degrades performance across multiple settings.
>
> We think if high-quality supervision combining accurate grounding with aligned CoT reasoning were available, SFT could potentially serve as a effective initialization. However, such data is currently scarce in the medical domain, which further motivates our RL-based formulation with simple data.
>
> |              | **PMC-VQA** | **PathVQA** | **SLAKE** | **VQA-RAD** | **Omni.** | **MMMU-Med** | **MedX.** | **Average** |
> |--------------|:-----------:|:-----------:|:---------:|:-----------:|:---------:|:------------:|:---------:|:-----------:|
> | Base         |    60.20    |    71.98    |   81.25   |    64.14    |   81.34   |     63.33    |   23.50   |    63.68    |
> | Base w/ SFT  |    49.90    |    66.30    |   62.02   |    66.93    |   59.19   |     68.00    |   23.30   |    56.52    |
> | ViToS        |    62.40    |    77.19    |   84.62   |    71.31    |   83.30   |     78.00    |   25.80   |    68.95    |
> | ViToS w/ SFT |    53.35    |    74.12    |   81.97   |    68.92    |   78.96   |     70.00    |   24.00   |    64.47    |

---

> > ### Author Rebuttal · Reviewer_R3Qs · 2026-04-03
> >
> > Thanks for the author's response. My concerns have been addressed, and I have decided to raise my rating to 4.

---

> > > ### Author Response · Authors · 2026-04-03
> > >
> > > We sincerely thank you for your positive feedback and for taking the time to review our responses. We are glad that all your concerns have been addressed.

---

### Official Review · Reviewer_kYWG · 2026-03-10

**Soundness:** 2
**Presentation:** 3
**Significance:** 3
**Originality:** 3
**Overall Recommendation:** 3
**Confidence:** 4

**Summary:**

This paper studies medical multimodal reasoning from the perspective that clinically relevant evidence in medical images is often sparse, while dense visual tokens may introduce distraction and unnecessary computation. To address this, the authors propose ViToS, a dual-stream reinforcement learning framework that couples a localization branch for identifying clinically relevant regions with a token-sparse reasoning branch that performs question answering after grounding-aware visual token pruning. The method further introduces token fusion to preserve contextual cues, a reward design that combines format, localization, and task accuracy signals, and a cross-feedback sequential optimization strategy to coordinate the two branches within a unified policy. Experiments on seven medical multimodal benchmarks and multiple medical VLM backbones show that the approach can reduce visual tokens while maintaining or improving reasoning performance, highlighting the promise of policy-guided sparse evidence selection for efficient medical VLMs.

**Compliance With Llm Reviewing Policy:**

Affirmed.

**Key Questions For Authors:**

1.	The paper introduces two sequential rollout stages during training, but it does not clearly quantify how much additional training overhead this serial design brings compared with a single-branch rollout.
2.	The background-token fusion process in Section 3.1.2 is not sufficiently clear. In particular, it is unclear how each background token is fused with the most semantically similar foreground token, and a more formal mathematical description would improve clarity.
3.	The definition of the task-specific accuracy reward is unclear, especially for open-ended textual answers. It is unclear how the method determines whether a prediction matches the ground truth. Is an LLM-as-a-judge strategy used?
4.	The advantage of DS-RL over a naive application of GTP to the base model is not fully convincing. DS-RL appears to bring only limited performance improvement compared to directly applying GTP, while introducing additional training costs. Furthermore, although Table 2 compares the combination of DS-RL and GTP with other token pruning methods, it fails to demonstrate the advantage of the DS-RL framework over these methods.
5.	The inference-efficiency comparison in Table 4 may not be fully fair, since the proposed method is evaluated with vLLM while other pruning methods do not seem to be compatible with the same backend. A comparison under a common non-vLLM inference setup would be more convincing.
6.	The proposed framework depends on curated training data, but the paper does not sufficiently describe the sources and construction process of the 19 object detection datasets. This raises concerns about possible overlap with the evaluation benchmarks and about the validity of the reported gains. In addition, it would be helpful to know how much improvement could be obtained by simply performing SFT on the same training data.

**Limitations:**

YES

**Strengths And Weaknesses:**

Strength：
The motivation of this paper is generally reasonable and well grounded. The proposed method addresses the redundancy of visual tokens in medical image reasoning, and the introduced ViToS framework enables the model to actively identify query-relevant visual tokens, avoiding the need for manually designed and potentially cumbersome rules. The method also achieves a certain degree of performance improvement while improving inference efficiency. In addition, the paper validates the effectiveness of combining the proposed approach with the GTP strategy across a wide range of benchmarks and different base models.
Weaknesses：
The main weaknesses of this paper lie in the fairness of the experimental design, as well as the lack of sufficient clarity in the description of the training data and the algorithmic process.

---

> ### Author Rebuttal · Authors · 2026-03-31
>
> We sincerely thank the reviewer for the positive and constructive feedback. We appreciate your recognition that the motivation of our work is well grounded. We are also glad that you find our ViToS framework effective in enabling the model to actively identify query-relevant visual tokens. Regarding your concerns, we provide our detailed responses below:
>
> > (Weakness 1 and Question 1) Additional training overhead.
>
> Due to space limitations, a detailed discussion of the training overhead **is provided in our response to `Reviewer tifr` (Weakness 2)**.
>
> > (Question 2) Lack of clarity in the background-token fusion process.
>
> Thank you. In our method, for each background token, we compute its cosine similarity with all foreground tokens and assign it to the most semantically similar foreground token via an argmax operation. The assigned background tokens are then grouped and fused with their corresponding foreground token through normalized aggregation. A formal description is already presented **in a pseudocode format (see Alg. 1 in Appendix Section B)**. We will enhance this detail in the final version of the main text.
>
> > (Question 3) Unclear definition of the task-specific accuracy reward.
>
> In our study, we **do not use any LLM-as-a-judge strategy**. Our experiments are conducted on closed-ended medical VQA benchmarks. In this setting, the model is required to output an answer in a fixed multiple-choice format (e.g., \\boxed{}), as specified in the prompt (see Appendix Sec. G). The predicted answer is then directly matched with the ground-truth choices to directly compute the accuracy.
>
> > (Question 4) Limited advantage of DS-RL over GTP.
>
> Thank you. We admit that our presentation in Table 2 is not clear enough and would like to clarify that DS-RL is not merely designed to improve performance over GTP, but to fundamentally transform how token pruning is realized in medical VLMs.
>
> Our proposed GTP operates as **an inference-time heuristic** that mitigates visual token redundancy by selecting grounded tokens. It thus **relies on twice rollouts and does not internalize** token-sparse reasoning within the model's ability.
>
> In contrast, DS-RL teaches the model to internalize the ability to focus on critical visual evidence. DS-RL **reformulates visual token pruning as a policy-guided evidence selection process and enables end-to-end training** of token-sparse reasoning. DS-RL thus helps to model to actively select the crucial visual tokens (a proximation to GTP). After DS-RL, the model achieves comparable or better performance with **a single rollout** (**3rd row in Table 2**). Moreover, DS-RL benefit to a better reasoning of the model by adapting it to the sparse evidences. DS-RL can **be seamlessly combined with proposed GTP** at inference, leading to further improvements over either GTP or DS-RL (**4th row in Table 2**).
>
> > (Question 5) Potential unfairness in inference-efficiency comparison due to different backends (vLLM vs non-vLLM).
>
> We have conducted experiments under a unified non-vLLM setup. Our approach still **achieves better efficiency compared to existing pruning methods (e.g., VisionZip, VScan, FastV, PDrop) when not using vLLM**, especially for larger models (e.g., 32B). Existing pruning methods incur increasing computational overhead when scaling to larger models, particularly in determining which tokens to prune, whereas our approach learns a more efficient policy after the DS-RL training.
>
> In line 322-324, we claim that "We attribute this efficiency advantage to the fact that our approach is still compatible with the vLLM inference framework." In addition, we will include clearer non-vLLM comparisons and refine the discussion in the final version to ensure a more convincing and fair evaluation.
> |     | **VisionZip** | **Vscan** | **FastV** | **PDrop** | **ViToS (non-vLLM)** | **ViToS (vLLM)** |
> |-----|:-------------:|:---------:|:---------:|:---------:|:--------------------:|:----------------:|
> | 7B  |       50      |     54    |     52    |     47    |          31          |        11        |
> | 32B |      172      |    184    |    171    |    202    |          81          |        53        |
>
> >(Question 6) Unclear data sources and lack of comparison with SFT on the same data.
>
> We thank the reviewer for raising these important concerns and we address them in two major aspects:
>
> - Regarding to data sources, 19 object detection datasets are collected from the Roboflow platform and curated into our used dataset. The construction process is described in ViTAR. These datasets are designed for grounding, which are ensured not to overlap with the evaluation VQA benchmarks.
> - Due to space limitations, a detailed discussion of the comparison with SFT is provided **in our response to `Reviewer R3Qs` (Question 1)**.

---

> > ### Author Rebuttal · Reviewer_kYWG · 2026-04-02
> >
> > I have read the response from the authors and thank them for their efforts to improve the quality of the manuscript. Since most of my concerns have been resolved, I would raise my score to 4 - weak accept. However, I still have some questions.
> > 1.	Although the proposed method is evaluated on closed-ended VQA benchmarks, the accuracy reward is calculated during the training stage on the curated training datasets. Are the training datasets also in a multiple-choice format? Can the authors provide some examples from the training datasets?
> > 2.	Besides, although the authors claim that the curated 19 object detection datasets come from the Roboflow platform, it is unclear which datasets were collected from this platform. Can the authors provide a short list of the 19 collected datasets to avoid potential dataset overlap.

---

> > > ### Author Response · Authors · 2026-04-03
> > >
> > > We sincerely thank the reviewer for the insightful follow-up questions and for acknowledging that our previous responses have addressed most of concerns. We particularly appreciate your willingness to consider raising your score. We appreciate your valuable suggestions, which help us further clarify important details of our work.
> > >
> > > > Q1. About training datasets
> > >
> > > The curated training datasets used in our method are constructed in a multiple-choice VQA format. Each sample consists of a question, a answer, the ground-truth answer, and the corresponding bounding box annotation for the relevant region.
> > >
> > > We provide **several representative examples below for clarification**:
> > >
> > > - Example Question 1: Is there a liver tumor in the image? A) Yes B) No. Ground Truth: A.  Bounding box: [0.7,0.61,0.87,0.8].
> > >
> > > - Example Question 2: Is there a nodule in the image? A) Yes, only in the left half B) Yes, only in the right half C) Yes, in both left and right halves D) No. Ground Truth: B.  Bounding box: [0.6,0.52,0.65,0.57].
> > > - Example Question 3: What is the type of tumor at image coordinates (0.7, 0.6)? A) Glioma B) Meningioma C) Pituitary D) Other normal area. Ground Truth: A. Bounding box: [0.65,0.58,0.74,0.67].
> > >
> > > We omit the visualization of images here for brevity. We will **make the curated training datasets publicly available** to ensure reproducibility and transparency.
> > >
> > > > Q2. About the 19 collected datasets
> > >
> > > The 19 object detection datasets used in our work are collected from the Roboflow platform, primarily based on the dataset introduced in Roboflow100-vl [1].
> > >
> > >
> > > To improve transparency, we include a concise list of the selected datasets:
> > > - https://universe.roboflow.com/okul-afczc/tumor2-ib7hn
> > > - https://universe.roboflow.com/college-fp3l1/lung-nodule-detection-2lebo
> > > - https://universe.roboflow.com/rf100-vl/canalstenosis-azjxm-cpkp
> > > - https://universe.roboflow.com/rf100-vl/crystal-clean-brain-tumors-mri-dataset-hzb2f-plsq
> > > - https://universe.roboflow.com/gradworkspace-zbl1g/mri-brain-tumors
> > > - https://universe.roboflow.com/new-workspace-qysn1/breastcancer-bl1ys
> > > - https://universe.roboflow.com/brain-tumor-xafk6/liver_tumor
> > > - https://universe.roboflow.com/science-fair-sluge/kidney-tumor-detection-golwd
> > > - https://universe.roboflow.com/rf100-vl/dentalai-i4clz-fsuo-ung2d
> > > - https://universe.roboflow.com/rf100-vl/inbreast-zzlbj-e5zj8-bzvi
> > > - https://universe.roboflow.com/rf100-vl/nih-xray-itazg-xeoi
> > > - https://universe.roboflow.com/rf100-vl/spinefrxnormalvindr-lt1cn-ryhy
> > > - https://universe.roboflow.com/rf100-vl/xray-2vqog-u6ggy-gqwl
> > > - https://universe.roboflow.com/rf100-vl/urine-analysis1-2lol7-onpk
> > > - https://universe.roboflow.com/colonnew/hera8k_t
> > > - https://universe.roboflow.com/rf100-vl/liver-disease-jyvvu-fash
> > > - https://universe.roboflow.com/technikum-f29gx/medical-imaging
> > > - https://universe.roboflow.com/glaucoma-mtzwc/glaucoma-0zetc
> > > - https://universe.roboflow.com/veri-etiketleme-tfqe1/ekg-train-2
> > >
> > > ---
> > > [1] Robicheaux P, Popov M, Madan A, et al. Roboflow100-vl: A multi-domain object detection benchmark for vision-language models[J]. arXiv preprint arXiv:2505.20612, 2025.
> > >
> > > ---
> > > **updates**：
> > >
> > > We hope that our latest clarifications satisfactorily answer your remaining questions.
> > >
> > > We look forward to your further feedback and would greatly appreciate any update to your evaluation if you find our responses sufficient. **We also noticed that the score in the system has not yet been updated, and we would be grateful if you could reflect your latest assessment when convenient.**
> > >
> > > Should there be any further questions or follow-ups, we would be happy to respond promptly. Thank you again for your time and thoughtful feedback.

---

### Official Review · Reviewer_tifr · 2026-03-10

**Soundness:** 3
**Presentation:** 3
**Significance:** 3
**Originality:** 3
**Overall Recommendation:** 4
**Confidence:** 3

**Summary:**

This paper focuses on a key challenge in medical multimodal reasoning, where critical visual evidence is sparse while the number of visual tokens is highly redundant. To address this issue, the authors propose a dual-stream reinforcement learning framework, ViToS, which uses the same policy model to perform both visual evidence localization and localization-guided token-sparse reasoning. The framework introduces grounding-aware token pruning and a serialized optimization strategy with cross-feedback to mitigate the coupling problem caused by sharing the policy across two branches. Experimental results on multiple medical VQA benchmarks and different vision–language models show that the proposed method can significantly improve performance while reducing the number of tokens and improving reasoning efficiency, demonstrating its effectiveness.

**Compliance With Llm Reviewing Policy:**

Affirmed.

**Final Justification:**

The authors’ response has addressed my concerns, and I will maintain my score.

**Key Questions For Authors:**

1.Lesions in medical images are often irregular in shape. Since the method relies on bounding boxes for localization, could the inability of boxes to accurately capture irregular lesions affect the reasoning performance?
2.The experiments mainly focus on the VQA task. Could the authors discuss whether the proposed framework can be extended to other tasks, such as medical report generation or cross-modal retrieval?

**Limitations:**

Yes

**Strengths And Weaknesses:**

Strengths:
1.The paper addresses the practical challenge of sparse key visual evidence but excessive token redundancy in medical multimodal reasoning. The motivation is clear and has practical application value.
2.The method integrates visual grounding and token-level reasoning into a unified reinforcement learning framework. The use of grounding-aware token pruning and cross-feedback rewards improves training stability, and the overall design is coherent.
3.Experiments conducted on multiple medical VQA benchmarks and different vision–language models demonstrate the effectiveness of the proposed approach.

Weaknesses:
1.The method relies heavily on accurate localization. If the localization is inaccurate, the fusion mechanism may not compensate for the loss of key information. In addition, the localization branch uses IoU-based rewards and requires ground-truth bounding boxes, which may increase the application cost in scenarios where annotations are scarce.
2.The overall implementation is relatively complex. The dual-branch architecture, cross-feedback mechanism, and serialized training strategy may significantly increase training time. The paper should further discuss the computational and training costs.

---

> ### Author Rebuttal · Authors · 2026-03-31
>
> We sincerely thank the reviewer for the constructive feedback. We appreciate your recognition of the practical motivation of our work. We are also grateful for your positive evaluation of method design, including the integration of visual grounding and token-level reasoning within a unified RL. Regarding your concerns, we provide our detailed responses below:
>
> > (Weakness 2) Computational and training costs.
>
> We provide a detailed breakdown of the training cost compared to a naive RL (single branch) baseline. As shown in our analysis, **the increase is modest (about 12\%) relative to the overall training cost, rising from 166.43s to 186.67s per step**. The additional overhead mainly comes from the rollout stage (39.41 vs. 20.73), which is reasonably expected since our dual-stream design requires twice rollouts. Importantly, other components-such as log probability computation for both the old policy and the reference model, actor updates, and auxiliary computations-**remain comparable across the two settings**, as we update only one branch while keeping the other frozen.
>
> |            | **Rollout** | **Old policy** | **Reference model** | **Actor updates** | **Auxiliary** | **All times** |
> |------------|:-----------:|:--------------:|:-------------------:|:-----------------:|:-------------:|:-------------:|
> | Vanilla RL |    20.73    |      32.70     |        26.29        |       83.21       |      3.50     |     166.43    |
> | Ours       |    39.41    |      33.05     |        26.10        |       84.60       |      3.52     |     186.67    |
>
> > (Weakness 1 \& Question 1) Limitations of box-based localization: irregular lesions, sensitivity to accuracy, annotation cost.
>
> Thank you, we would like to address this concern from three detailed aspects:
>
> - Irregular lesions.
> The bounding boxes are derived following standard object detection practices. Thus irregular lesion regions **are tightly enclosed using their maximal coordinate extents**, ensuring that most relevant regions are covered despite shape irregularities.
>
> - Sensitivity to localization accuracy.
> Our method is explicitly designed to be robust to imperfect localization. Instead of naive image cropping, we purposely **perform the token-level pruning**. Empirically, merely discarding redundant visual information is insufficient for robust medical visual reasoning (see Fig. 8). To address this, we **introduce a token fusion mechanism** that integrates information from non-critical regions into foreground visual tokens. This key design preserves the semantic saliency while maintaining global contextual cues. As a result, we observe stable performance gains across datasets after token fusion (see Appendix Sec. A.3).
>
> - Annotation cost.
> Bounding box annotations are widely available in the medical imaging community. They can also be efficiently derived from existing segmentation masks with minimal overhead. In our work, we **rely on publicly available detection datasets and make this requirement practical and scalable**.
>
> > (Question 2) Limited evaluation beyond VQA tasks.
>
> To extend the scope of our framework, we have conducted additional experiments on medical report generation benchmarks, including IU-Xray, CheXpert-Plus, and MIMIC-CXR. As shown in the results, our method **improves performance across multiple metrics (e.g., ROUGE-L and RaTE) compared to the baseline**, demonstrating that the benefits of grounding-aware token pruning and token-sparse reasoning extend beyond VQA to more generative tasks. In addition, we observe **gains across different model backbones and scales**, including Lingshu and HuatuoGPT-Vision under 7B and 32B settings.
> |                     | IU-Xray (ROUGE-L) | IU-Xray (RaTE) | CheXpert-Plus (ROUGE-L) | CheXpert-Plus (ReTE) | MIMIC-CXR (ROUGE-L) | MIMIC-CXR(RaTE)| Average (ROUGE-L) | Average (RaTE)  |
> |---------------------|-------------------|----------------|-------------------------|----------------------|---------------------|-------|---------|-------|
> | Lingshu-7B          | 40.66             | 58.65          | 25.89                   | 42.57                | 34.20               | 52.80 | 33.58   | 51.34 |
> | + ViToS             | 45.57             | 65.02          | 23.92                   | 48.52                | 32.93               | 56.08 | 34.14   | 56.54 |
> | Lingshu-32B         | 42.78             | 63.01          | 25.91                   | 44.18                | 34.23               | 52.58 | 34.30   | 53.26 |
> | + ViToS             | 42.59             | 65.35          | 27.55                   | 46.69                | 33.85               | 56.21 | 34.66   | 56.08 |
> | HuatuoGPT-Vision-7B | 18.43             | 44.57          | 20.98                   | 42.38                | 21.20               | 45.89 | 20.20   | 44.28 |
> | + ViToS             | 28.61             | 53.03          | 23.61                   | 43.23                | 25.24               | 48.22 | 25.82   | 48.16 |

---

> > ### Author Rebuttal · Reviewer_tifr · 2026-04-01
> >
> > The authors’ response has addressed my concerns, and I will maintain my score.

---

> > > ### Author Response · Authors · 2026-04-02
> > >
> > > We sincerely thank you for your thoughtful review and for recognizing that our responses have adequately addressed your concerns. We greatly appreciate your time and constructive feedback.
> > >
> > > If there are any remaining questions or points that you would like us to further clarify, we would be very happy to continue the discussion.
> > >
> > > If you feel that our revisions have sufficiently improved the paper, we would be grateful if you would consider adjusting your score accordingly. Thanks again.

---

### Official Review · Reviewer_4Y3f · 2026-03-13

**Soundness:** 3
**Presentation:** 3
**Significance:** 2
**Originality:** 2
**Overall Recommendation:** 4
**Confidence:** 4

**Summary:**

This paper proposed a method to combine both sparse visual evidence selection and token-sparse reasoning by leveraging dual stream reinforcement learning method. To make the two branches work together, they introduce a cross-feedback and sequential optimization strategy, so the model first learns where to look and then learns how to reason efficiently from the selected evidence. Experiments on seven medical VQA and multimodal benchmarks show that ViToS consistently improves accuracy over strong medical VLM baselines while also reducing inference cost.

**Compliance With Llm Reviewing Policy:**

Affirmed.

**Final Justification:**

This paper have revealed an interesting intuition for medical VLM and show that we can use RL method to pinpoint the sparse token that  is important to the image understanding. The contribution is substansious. Therefore I will keep my score.

**Key Questions For Authors:**

A useful addition would be a dedicated comparison against simple image-cropping baselines. Right now, the paper argues that grounding-aware token pruning is better than directly cropping the image, but this point is not validated as thoroughly as the main pruning and RL comparisons. Though the technical contribution is not ground-breaking but is effective and sound. Considering the majority of the reviewers give a positive score and their position also reflect mine. I will keep my score.

**Limitations:**

yes

**Strengths And Weaknesses:**

strength
1. This paper has shown that the idea of first select sparse evidence tokens and then reasoning from these selected is actually more beneficial for medical scenairo. The authors argued that this following the intuition in medical scenario that sometimes the important information in medical images are quite sparse.
2. The design of cross-feedbacka and sequential optimization is quite important and increase the training process stability. Rather than mix the localization and reasoning signal together at the beginning. The proposed method using a sequential optimizaiton method to make the training proces much more stable.
3. The token fusion method is also a good design to smooth the localization training objective. It is like a window stacking method that focus on the detail in the local window but do not give up the information near the boundary.

weakness
1. Though the sequential optimization method is quite useful in this scenario. But this is a commonly accepted traning strategy in curricum learning and sequential RL learning. It is not that novel.
2. If the image size is large, the localization stream will be very hard to train. But these kind of image are quite common in medical scenairo. (PET, WSI, etc.) I think this method may suffer in localization action space on these modality of data.

---

> ### Author Rebuttal · Authors · 2026-03-31
>
> We sincerely appreciate the reviewer’s thorough and encouraging feedback. Thank you for recognizing the motivation of our work and for summarizing our approach clearly. We are also grateful for your positive comments on the optimization design, as well as the token fusion mechanism. Regarding your concerns, we provide our detailed responses below:
>
> > (Weakness 1) Novelty of the sequential optimization strategy.
>
> We would like to clarify that our contribution goes beyond standard curriculum learning or sequential RL training. The key novelty lies in **reformulating visual token pruning as a policy-guided evidence selection** under end-to-end training for token-sparse medical reasoning. Building upon this, we introduce **a dual-stream RL framework using a united policy model** with two cascaded branches. Additionally, we propose the **cross-feedback sequential optimization** to avoid gradient conflict and facilitate the convergence of the shared policy model.
>
> > (Weakness 2) Scalability of the localization stream to high-resolution medical images (e.g., WSI, PET).
>
> We agree that scaling to extreme-resolution medical images (WSI and PET scans) can introduce a larger localization action space as a challenge for policy learning. In this study, we follow the standard set-up from related works by adopting standard-resolution medical image, where ViToS can be extensively developed and evaluated. Our study is not specifically designed for such extreme-resolution settings. Extending our framework to handle extreme-resolution inputs (e.g., via hierarchical strategies for localization) is an important direction for future work. We will clarify this point in the final version with an additional limitation paragraph.
>
> > (Question 1) Insufficient comparison with simple image-cropping baselines.
>
> We have added experiments comparing our method with both image-level cropping and token-level cropping baselines. As shown in the attached results, naive image cropping and token cropping only achieve limited improvements over the base model, while **our ViToS achieves remarkable performance gain**. Notably, cropping methods exhibit inconsistent behavior across datasets with degrading performance (e.g., PMC-VQA, SLAKE), whereas our method consistently improves results. We further compare with general-domain methods that rely on direct image cropping in reasoning, such as DeepEyes [1] and PixelReasoner [2]. We observe that these general VLMs struggle to perform well on medical images, highlighting **the importance of domain-specific GTP rather than coarse image cropping**. To illustrate, this difference stems from the fact that cropping performs hard spatial truncation and may discard clinically relevant contextual information. In contrast, our method conducts grounding-aware, token-level evidence selection and preserves contextual cues via token fusion, leading to more stable and effective reasoning. We will include this analysis in the final version.
>
> |               | PMC-VQA | PathVQA | SLAKE | VQA-RAD | Omni. | MMMU-Med | MedX. | Average |
> |---------------|:-------:|:-------:|:-----:|:-------:|:-----:|:--------:|:-----:|:-------:|
> | Base          |   60.2  |   72.0  |  81.3 |   64.1  |  81.3 |   63.3   |  23.5 |   63.7  |
> | Image Crop    |   53.6  |   74.8  |  74.0 |   70.5  |  80.4 |   70.7   |  25.1 |   64.2  |
> | Token Crop    |   54.8  |   74.2  |  76.9 |   70.5  |  81.8 |   70.0   |  25.2 |   64.8  |
> | DeepEyes   |    -    |   52.9  |  68.2 |   65.9  |  64.8 |   57.8   |  23.6 |    -    |
> | PixelReasoner |    -    |   52.6  |  67.3 |   66.0  |  64.9 |   58.0   |  23.5 |    -    |
> | ViToS (Ours)  |   62.4  |   77.2  |  84.6 |   71.3  |  83.3 |   78.0   |  25.8 |   68.9  |
> ---
> References:\
> [1] Zheng Z, Yang M, Hong J, et al. Deepeyes: Incentivizing" thinking with images" via reinforcement learning[J]. arXiv preprint arXiv:2505.14362, 2025. \
> [2] Wang H, Su A, Ren W, et al. Pixel reasoner: Incentivizing pixel-space reasoning with curiosity-driven reinforcement learning[J]. arXiv preprint arXiv:2505.15966, 2025.

---

> > ### Author Rebuttal · Reviewer_4Y3f · 2026-04-04
> >
> > I have no further question and will maintain my score.

---

> > > ### Author Response · Authors · 2026-04-04
> > >
> > > Thank you so much for your positive response and for acknowledging that our responses have resolved your concerns. We appreciate the time you have dedicated to reviewing our work.

---

### Decision · Program_Chairs · 2026-04-30

**Decision:**

Accept (regular)

**Comment:**

The final reviewer scores are 4/4/4/3. However, all reviewers expressed an overall positive attitude toward the paper. Notably, the only remaining negative score appears to be solely because the reviewer kYWG forgot to update the rating after explicitly indicating in the latest comment that they would raise it to 4 (Weak Accept). Given that all reviewers were ultimately positive about the submission, my recommendation is Weak Accept.